# Physics-informed deep generative learning for quantitative assessment of the retina

Emmeline E. Brown[1,2], Andrew A. Guy [1,3], Natalie A. Holroyd [1],
Paul W. Sweeney [4], Lucie Gourmet[1], Hannah Coleman[1], Claire Walsh [1,5],
Athina E. Markaki [3], Rebecca Shipley [1,5], Ranjan Rajendram [2,6] &
Simon Walker-Samuel [1] ✉

Disruption of retinal vasculature is linked to various diseases, including diabetic retinopathy and macular degeneration, leading to vision loss. We present here a novel algorithmic approach that generates highly realistic digital models of human retinal blood vessels, based on established biophysical principles, including fully-connected arterial and venous trees with a single inlet and outlet. This approach, using physics-informed generative adversarial networks (PI-GAN), enables the segmentation and reconstruction of blood vessel networks with no human input and which out-performs human labelling. Segmentation of DRIVE and STARE retina photograph datasets provided near state-of-the-art vessel segmentation, with training on only a small ($n = 100$) simulated dataset. Our findings highlight the potential of PI-GAN for accurate retinal vasculature characterization, with implications for improving early disease detection, monitoring disease progression, and improving patient care.

Disruption of retinal vasculature is associated with a range of diseases which can result in loss of vision, including diabetic retinopathy (DR)[1] and macular degeneration[2]. It is also increasingly recognized that retinal vasculature can indicate the presence of systemic pathology, such as vascular dementia[3] and cardiovascular disease[4]. Automated methods to characterize and quantify changes in retinal vasculature from clinical imaging data therefore offer substantial promise for high-throughput, early detection of disease[5], which is critically required to meet the increasing incidence of retinal disease, potentially alongside other vascular diseases, and their associated burden on healthcare systems[6].

Much attention has been placed on supervised deep learning in this regard, where deep neural networks are trained to categorise images according to diagnosis or identify the location of features of interest[7]. Supervised learning, particularly with U-net architectures[8], first rose to prominence in retinal image analysis for segmenting retinal layers in optical coherence tomography (OCT) data[9], alongside

blood vessel segmentation in retinal photographs[10,11]. A significant limitation to this type of approach is the lack of high-quality, manually-labelled image data in sufficient quantities to enable accurate and generalisable predictions to be made[12]. Novel deep learning architectures and hyperparameter selection are often employed to mitigate issues of smaller or lower quality datasets[13,14], though there is consensus that the input data is of primary importance. The problem is particularly acute for the detection of blood vessels, in which manual labelling is highly time-consuming, generally limited to two-dimensional (2D) projections, confined to larger vessels only, and generally does not distinguish between arteries and veins[15]. Manual segmentation of a single 2D retinal image can take multiple hours[16]. Inter and intra grader variability can also be significant within the segmentation process[17,18]. Most segmentation studies have been conducted in 2D retinal fundus photographs using public datasets[12,19,20]. Previous studies have successfully used GAN based segmentation to segment retinal fundus[21] and OCT-A[22] images, without the need to

[1]Centre for Computational Medicine, University College London, London, UK. [2]Moorfields Eye Hospital, London, UK. [3]Department of Engineering, University of Cambridge, Cambridge, UK. [4]Cancer Research UK Cambridge Institute, University of Cambridge, Cambridge, UK. [5]Department of Mechanical Engineering, University College London, London, UK. [6]Institute of Ophthalmology, University College London, London, UK. ✉e-mail: simon.walkersamuel@ucl.ac.uk

create a generalisable model. Synthetic data approaches have also been deployed for blood vasculature on experimental imaging datasets (mesoscopic photoacoustic imaging)[23]. Approaches that utilise and extend these techniques for application on high resolution wide-field images are urgently needed to enable the robust translation of deep learning into the clinic.

To address these challenges, we describe here a novel set of algorithms that can generate highly realistic digital models of human retinal blood vessels, using established biophysical principles and unsupervised deep learning. Our biophysical models capture the complex structure of retinal vasculature, with interconnecting arterial and venous trees that are inherently three-dimensional, multi-scale and fully inter-connected via a capillary bed. They also feature dedicated macula and optic disc features. The central biophysical principles we draw on are (1) Murray's Law, in which vessel diameters, branching distances and branching angles are optimised to form a balance between pumping power and blood volume and minimize resistance to flow[24]; and (2) fluid dynamics to model blood flow and vascular exchange. The latter is made possible by our synthetic networks containing fully-connected arterial and venous trees with a single inlet and outlet (the central retinal artery and vein), allowing blood flow and contrast agent delivery (e.g. fluorescein) data to be

simulated with minimal assumptions in regard to network boundary conditions. The simulation of fluorescein angiography (FA) data, in which the time- and flow-dependent delivery of a contrast agent enables the characterisation of retinal vessels, is of particular interest as it facilitates the validation of temporal dynamics. The dye is also poorly tolerated by patients which reduces its usage clinically; simulated data could therefore be a valuable alternative[25,26].

In this work, we investigate whether, through the use of generative deep learning, our biophysics-informed vascular network models can be used to infer information from real-world retinal images, such as the segmentation and reconstruction of blood vessel networks, without the need to perform any manual labelling, in an approach termed physics-informed generative adversarial learning (PI-GAN)[27]. An overview of our framework is provided in Fig. 1. Generative adversarial networks (GANs) incorporating cycle-consistency have previously been used for medical imaging domain machine learning tasks such as chest MRI to X-ray CT transformation[28], PET image denoising[29], and artefact reduction in fundus photography[30]. Likewise Menten et al used the space colonisation algorithm to generate macular blood vessel images, which they coupled with deep learning[31].

We demonstrate here the ability of our retinal simulation framework to accurately simulate real-world retinal vasculature, including

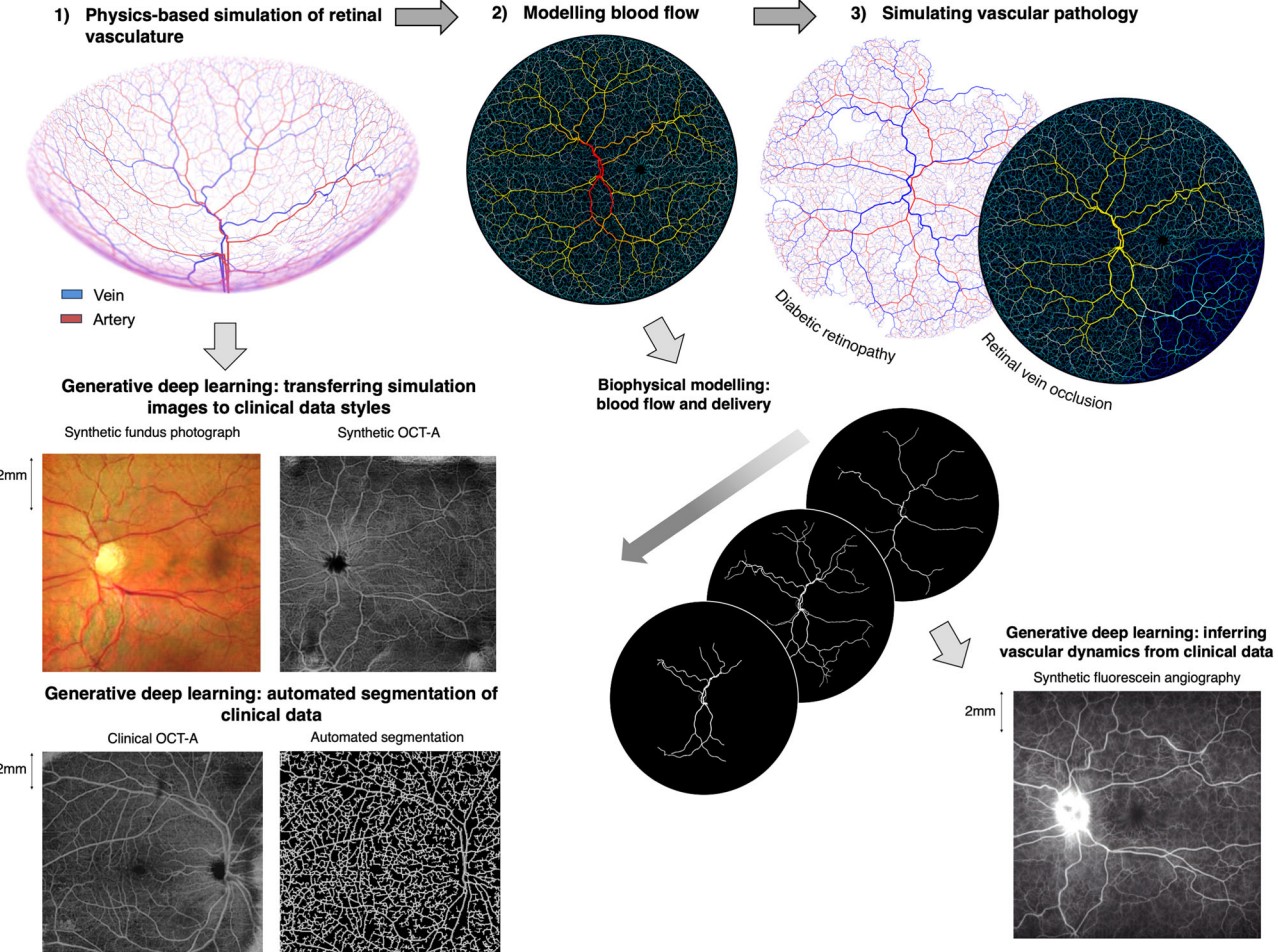

**Fig. 1 | Schematic overview showing the physics-informed generative adversarial learning (PI-GAN) framework developed in this study.** Retinal blood vessel networks, featuring arterial and venous trees connected by a capillary bed, and special treatment of macula and optic disc features, were simulated using space filling growth algorithms based on Murray's law. Blood flow and fluorescein delivery were simulated in synthetic vascular networks, using one-dimensional Poiseuille flow. By combining this with cycle-consistent, physics-informed deep generative learning, vessel simulations were converted into synthetic medical image data (fundus photography, Optical coherence tomography angiography (OCT-A) and FA), and the same trained networks used to detect blood vessels in clinical images. Image scale bar is given in millimeters (mm).

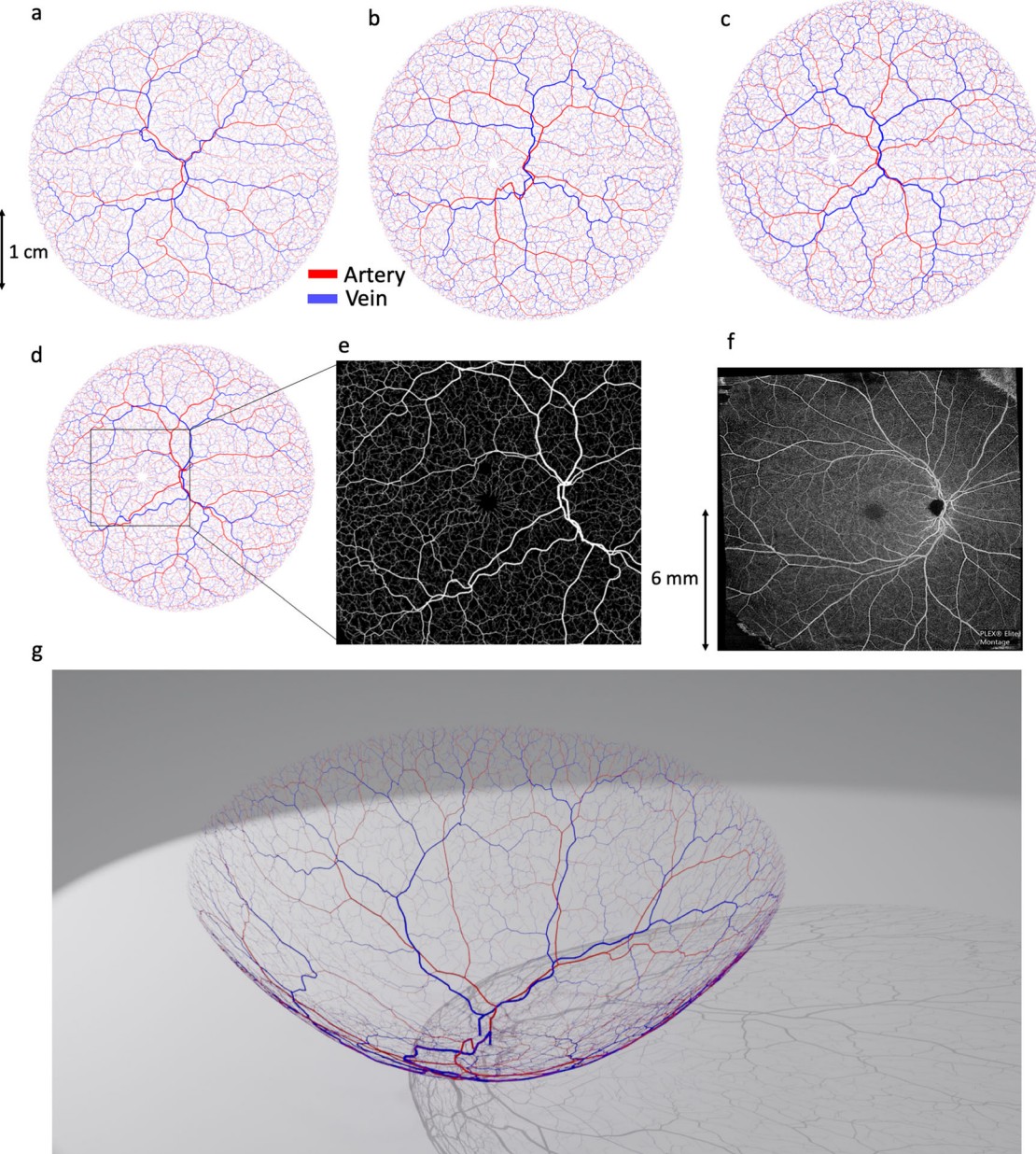

**Fig. 2 | Procedural generation of retinal vasculature using constrained constructive optimisation and lattice sequence vascularisation. a–c** Examples of synthetic retinal vascular networks, featuring arterial (red) and venous (blue) trees, and with geometry optimised according to Murray's law. Each simulation run used a different set of physiological parameter values, randomly sampled from the distributions defined in Supplemental Table 1. Scale bar is given in centimeters (cm). **d–f** A synthetic retina (**d**) with a 12 × 12 mm region surrounding the optic disc and macula (**e**) compared with a real OCT-A image (**f**). **g** A simulated vascular network projected onto three-dimensional surface (scale bar in millimeters (mm)).

blood flow, and model the presentation of two common vascular pathologies: DR and retinal vein occlusion (RVO). Moreover, we show that our PI-GAN workflow allows retinal vasculature to be segmented without any human manual labelling, and which outperforms state-of-the-art supervised learning approaches. This therefore offers numerous opportunities for improved detection and quantification of retinal disease in clinical ophthalmology.

## Results

### Procedural modelling of retinal vasculature

Retinal vascular networks were simulated in multiple, linked steps, using a combination of algorithms that draw on the known geometry and biophysics of retinal vasculature. In total, our procedural model of retinal vasculature contained 26 parameters (Supplementary Table 1), each of which were randomly sampled to simulate the broad range of retinal geometries occurring in the population (Fig. 2a–c)[32,33].

Networks were seeded using a Lindenmayer-system (L-system)[34], in which initial central retinal artery and vein segments were positioned at the location of an optic disc and iteratively branched within a plane. The first arterial and venous segment radii were 135 ± 15 μm and 151 ± 15 μm, respectively[35]. Branching was performed asymmetrically to create characteristically large vessels surrounding the macula, with smaller branches reaching towards the periphery, as observed in retinal images[35].

Seeding L-systems were then grown to the edge of the retina using a variant of constrained constructive optimisation (CCO)[36–40]. This step transformed L-system networks into realistic, space-filling networks with geometries defined by Murrays law[24] (exponent of 2.4 ± 0.11[41]),

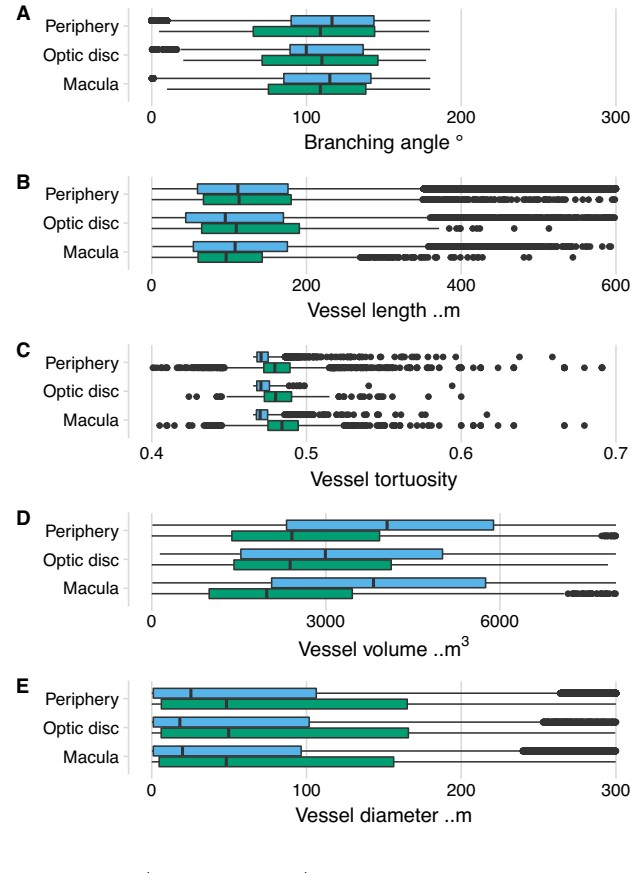

**Fig. 3 | Comparison of retinal vascular geometry distributions visualised with bar plots between manually segmented networks from OCT-A data (volunteers not ascertained for disease status)** $n = 19$ **images and simulated networks,** $n = 100$. The box plot displays the following summary statistics: median, lower and upper hinges (which correspond to the first and third quartiles (the 25th and 75th percentiles)), and upper and lower whiskers (the upper whisker extends from the hinge to the largest value no further than 1.5 x interquartile range (IQR) from the hinge), and the lower whisker extends from the hinge to the smallest value at most 1.5 x IQR. No replicates were included. **A** Branching angle, (**B**) vessel length (micrometers, μm) (length of the segment connecting two branch points in a graph vessel network), (**C**) vessel tortuosity (distance between vessel branching points divided by the Euclidean distance between them), (**D**) vessel volume (square micrometers, μm²) and (**E**) vessel diameter (micrometers, μm), in three regions: macula (5.5 mm diameter circular area centred on the fovea[45]), optic disc (3.6 mm diameter area centred on the optic disc centre[46]), and periphery (all vessels outside those regions). A two-sided Analysis of variance (ANOVA) was used to assess differences in these metrics by retina region (optic disc, macula, and periphery) and by status (healthy control and simulated network) (Supplementary Table 2) with eye (right OD/ left OS), participant sex, and OCT-A imaging scan pattern used as covariates.

whilst retaining the realistic macroscopic branching geometry imposed by the L-system seeding (Fig. 2d–f). A final growth step was incorporated to create the characteristic branching pattern of the macula, with radial alignment of arterioles and venules, greater relative vascular flow density (between 1.5 and 2.0 times the perfusion fraction) and a central avascular fovea.

Following growth, we augmented vessels with sinusoidal displacements to replicate the tortuous vasculature commonly observed in human retinas, with a greater displacement imposed on veins. A continuous capillary bed was generated using either (1) a 2D Voronoi algorithm that arterial and venous endpoints were connected to[42] or (2) a 2D space colonization algorithm[43]. Following simulation within a

2D plane, vessels were projected onto a hemispherical mesh (diameter 23–25 mm) featuring macula and optic disc structures generated using a mixed Gaussian profile[44] (Fig. 2, Supplementary Fig. 1, Supplementary Video 1).

## Comparison of synthetic networks with real-world networks

Our set of retinal network growth algorithms is designed to provide an authentic replication of real retinal vasculature, by following established biophysical principles. To quantitatively evaluate the accuracy of these synthetic networks, we manually labelled all visible blood vessels in 19 optical coherence tomography angiography (OCT-A) image datasets, using in-house software. This included differentiating arteries and veins (A-V) in a subset of images ($n = 5$), using retinal photographs as a reference for determining A-V status. Vessel branching angle, inter-branch length, tortuosity, and radius were measured in three regions: the macula, the vessels surrounding the optic disc, and the periphery. The macula was defined as a 5.5 mm diameter circular area centred on the fovea, based on measurements referenced in Remington and Goodwin[45]. The vessels surrounding the optic disc were labelled as a 3.6 mm diameter centred at the optic disc, due to mean vertical and horizontal diameters of the optic disc reported as 1.88 and 1.77 mm respectively[46]. Vessels outside these regions were defined as 'peripheral'. 100 synthetic retinal networks were initially created, with parameter values randomly drawn from the ranges shown in Supplementary Table 1.

According to ANOVA analysis, all geometric parameters associated with synthetic blood vessel networks did not reach the level of statistical significance compared to those measured in controls using manual segmentation of OCT-A images (branching angle, $p = 0.824$; vessel length, $p = 0.177$; vessel tortuosity, $p = 0.095$; vessel network volume, $p = 0.061$; vessel diameter, $p = 0.593$) (Fig. 3, Supplementary Table 2).

## Simulating retinal blood flow and validation using FA

We have previously developed a mathematical framework for simulating blood flow in three dimensional vascular networks, which uses one-dimensional Poiseuille flow[47]. Our simulated retinal networks are ideally suited for this framework, having just one arterial input and one venous outlet meaning that pressure boundary conditions can be easily specified. Setting arterial pressure by sampling from a normal distribution parameterised by mean = 56.2 mmHg, s.d. = 14.0 mmHg[48] and similarly for venous pressure with mean = 20.0 mmHg and s.d. = 10.0 mmHg[48] gave an average total retinal flow prediction of $34.4 \pm 1.8$ μL/min, which is slightly lower, but still in good agreement with reports in the literature from healthy retinas (for example, $45.6 \pm 3.8$ μL/min[48] $44 \pm 13$ μL/min[49] 50.7 μL/min[49] (Fig. 4a, b)).

To further evaluate these flow results, we next performed a simulation of retinal fluorescein delivery. FA is used in ophthalmology for diagnosis of macular oedema, macular degeneration, RVO, DR, and other diseases[26,50]. Fluorescein is injected as a bolus into the median cubital vein, and 10–15 s later appears in the choroidal vasculature at the rear of the eye[25]. Within 2 s of this, fluorescein appears in the anterior arteries and arterioles, and a further 2 s later by partial filling of venules and veins, followed by total filling and recirculation.

We simulated the systemic pharmacokinetics of fluorescein using literature data[4,51], with the passage of fluorescein modelled as two displaced Gaussian functions to model the first and second passes, and an exponential washout term corresponding to systemic extraction (Supplementary Fig. 2). This time course was propagated through our synthetic retinal networks by partitioning by flow at branch points and delaying according to cumulative velocities. The delay between arterial and venous filling with fluorescein, across 1000 simulation runs was $7.3 \pm 0.7$ s, which is in keeping with timings described in clinical data[25]. Visual inspection of fluorescein delivery also revealed a good accordance with clinical delivery profiles (Fig. 4c–h).

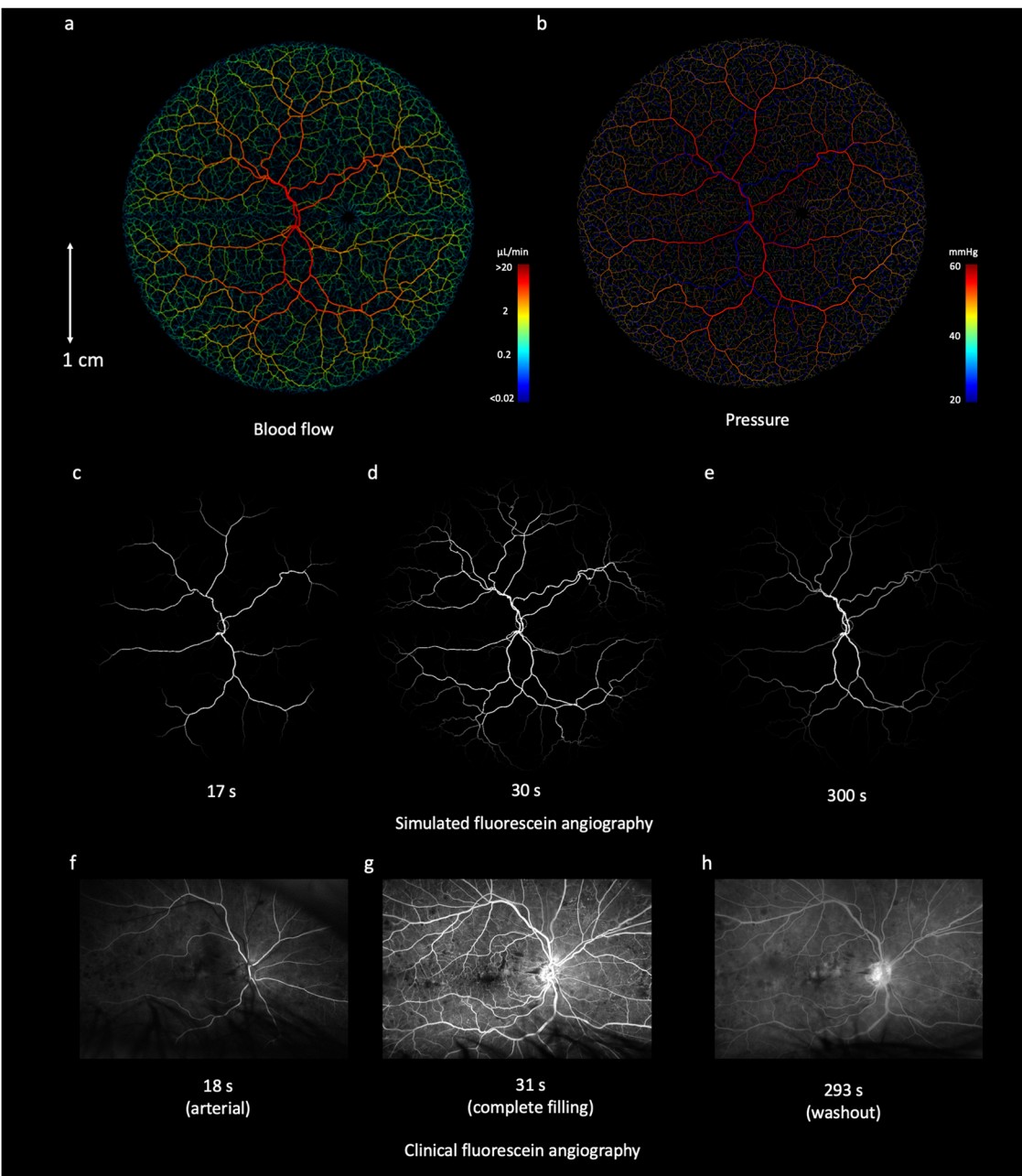

**Fig. 4 | Blood flow simulations in an example biophysical model of a retinal blood vessel network, with scale bar provided in centimeters (cm). a** Blood flow (microlitre per minute, µL/min) and (**b**) vascular pressure (millimeter of mercury, mm Hg) were simulated using Poiseuille flow, with inlet arterial pressure and outlet venous pressure fixed at 56.2 ± 14.0 and 20.0 ± 10.0 mmHg, respectively. **c**–**e** Simulated delivery of fluorescein at 17 s (arterial phase), 30 s (venous phase) and 600 s (recirculation), with clinical fluorescein images (registered to the same coordinate space) shown in (**f**–**h**) for comparison.

## Simulating retinal pathology

Given the physiologically-realistic results provided by our flow models, we next sought to perturb our simulated networks to examine the effect of pathological changes. As a first demonstration, we simulated the effect of RVO. A random location of artery-vein crossover on a large retinal vein was reduced in diameter by 80%. Blood flow within the network was recalculated, revealing a large region of hypoperfusion, as expected. This strongly reflected the presentation of RVO found in clinical FA data (Fig. 5) and induced a regional reduction in blood flow of 9.8 µL/min in the vessels immediately downstream of the occlusion.

Next, we constructed a simple model of DR[52–54], in which arterioles with a radius <35 µm were randomly selected and occluded, and the resultant change in network flow calculated. All vessels that become non-perfused, either up- or downstream of the occluded vessel, were removed from the network, creating regions of ischaemia, with occasional surviving vessels passing through (Fig. 5c, d, Supplementary Fig. 3). Occlusions were simulated in batches of 5, initially from the periphery (>1 cm from the macula centre), and then at decreasing minimum distances from the macula, as typically found in the clinical presentation of DR.

Both our retinal occlusion model and DR model produced images that were highly reminiscent of clinical images of both pathologies (Fig. 5), with loss of flow in downstream vessels in our RVO model and loss of perfusion and regions of ischaemia in the DR model.

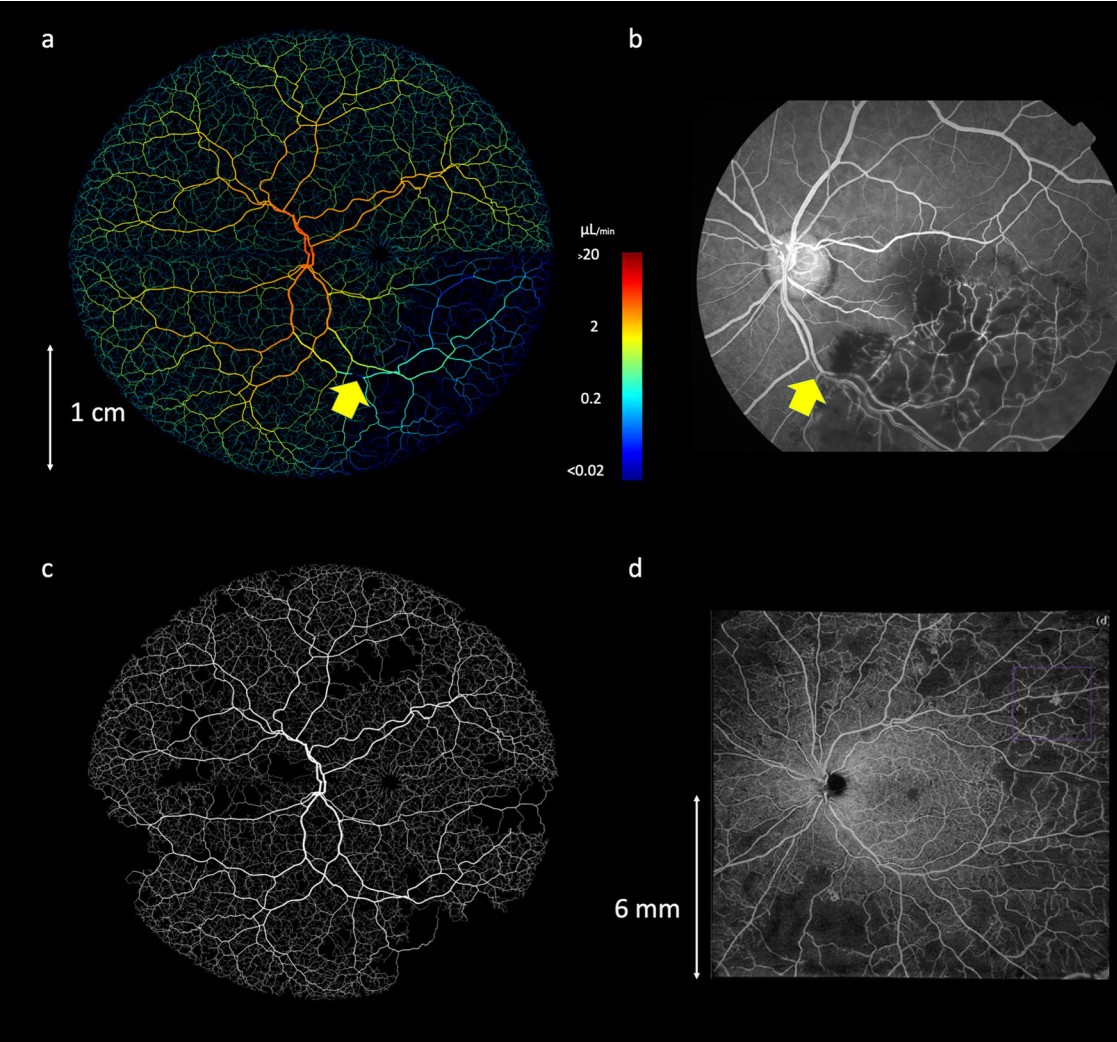

**Fig. 5 | Simulating realistic whole-retina pathology. a** An example of RVO simulation and loss of flow in downstream vessels. The yellow arrow shows the location of an imposed 80% decrease in vein diameter. The scale bar is in centimeters (cm) (**b**) An FA image of retinal occlusion, revealing a similar pattern of perfusion loss as simulated in (**a**). Colour bar unit is microliters per minute (**c**) The onset of DR, simulated by inhibiting flow in randomly-selected peripheral arterioles. **d** An OCT-A image of a retina exhibiting stage 4 DR, evidenced by extensive loss of perfusion in vessels and regions of ischaemia. The scale bar is shown in millimeters (mm).

## Generating synthetic clinical ophthalmology data with deep learning

Our next challenge was to use deep learning to define a mapping between our biophysical vascular model and clinical ophthalmology data (and vice versa). For this we used cycle-consistent generative adversarial networks that enabled the translation of image texture and style between image domains[55]. We undertook this for three clinical imaging modalities: OCT-A, retinal photographs and FA.

We embedded our synthetic retinas in three-dimensional grids with axial and lateral resolutions of 6.3 μm and 21 μm, respectively, to match our clinical OCT-A data. We then trained three cycle-consistent GANs on these synthetic retinas, with each GAN mapping the conversion between the synthetic images and a different imaging modality (Supplementary Fig. 4). 590 retinal photographs, 43 OCT-A en-face images and 570 FA images were used in training for this purpose. PI-GAN enabled the geometry of source images (simulations- synthetic space) to be translated into a target style (retinal photographs- retinal photograph space, OCT-A- OCT-A space, and FA- FA- space). As can be seen in Fig. 6a, following 400 training epochs, the pattern of synthetic vasculature was realistically transferred into the style of each target image. The Frechet Inception Distance (FID)[56] was 6.95 for retinal photographs, 5.17 for FA and 3.06 for OCT-A en-face images, indicating a small distance between feature vectors for real and synthetic images. FID calculated between two sets of real images from the same domain was used as a baseline (Supplementary Table 3).

This process generated authentic-looking retinal image data with matched, fully specified ground truth blood vessels. However, cycle consistency also allows the reverse operation: to generate simulation data from clinical images (Fig. 6b). This enabled blood vessel networks to be segmented from OCT-A images and compared with manual segmentations of the same data (Fig. 7). For segmentation of OCT-A data we trained PI-GAN with a synthetic dataset instance featuring smaller 'elusive' vessels[57] and capillary level vasculature. Visual inspection of PI-GAN segmentations revealed many more small 'elusive' vessels than represented within our manually-segmented images, arguably providing superior segmentation accuracy than the manual 'gold standard'. Accordingly, the mean Dice score for OCT-A images was low (mean 0.35, s.d. 0.12 (2.s.f)), but the sensitivity (the percentage of pixels labelled as vessel in the manual segmentation that were also identified as vessel by PI-GAN) was high (87.1% (s.d. 1.20)), showing that PI-GAN is able to accurately label almost all of the vessels identified by human operators (Fig. 7g–j).

a ) Forward direction

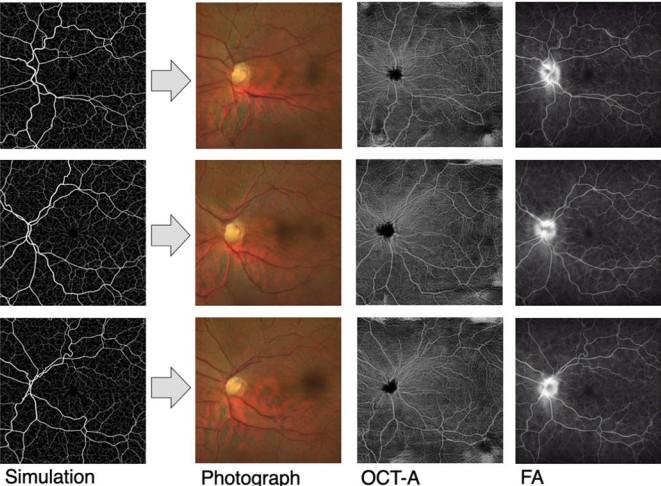

b ) Reverse direction

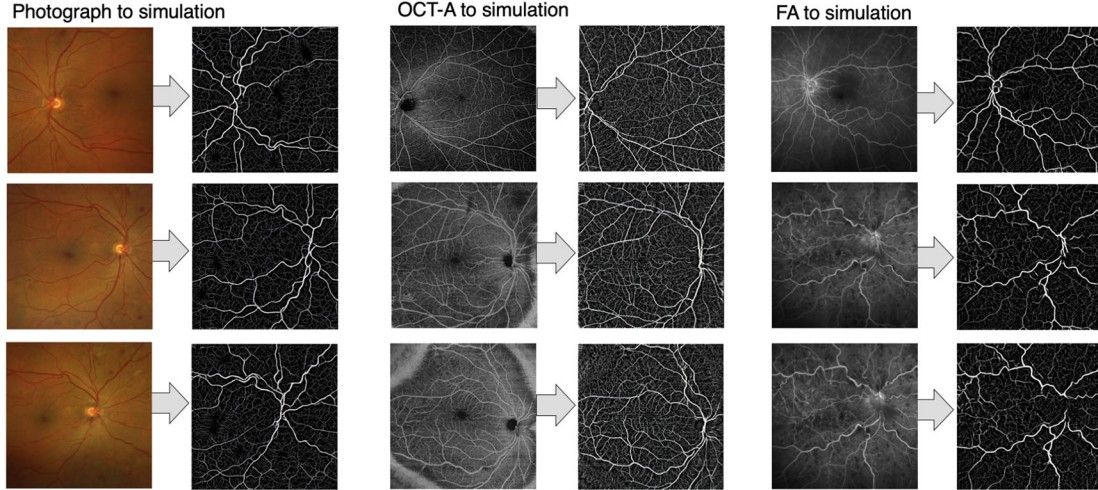

**Fig. 6 | Generation of multi-modality retinal images from biophysical simulations, using physics-informed generative adversarial networks. a** Direction 1 involves conversion of synthetic space (simulated network) into retinal photograph space, Optical coherence tomography angiography (OCT-A) space and fluorescein angiography (FA) space. **b** Direction 2 involves conversion of real retinal images (retinal photograph, OCT-A, and FA spaces) into fully connected networks/segmented data (synthetic space).

To further investigate the accuracy of the PI-GAN segmentation, we evaluated PI-GAN on two publicly available retinal photograph data sets with corresponding manual segmentations (STARE[19] and DRIVE[20]). PI-GAN was trained here using a synthetic dataset filtered such that the smallest vessel diameter corresponded to the resolution of retinal photography (taken to be 100 μm). Contrasting the widefield (130° and 200°) retinal photograph images analyzed in this study, these public datasets were acquired with a smaller 45 degree FOV, and are widely used in benchmarking vessel segmentation. Dice scores indicated a high accordance between manual segmentations and PI-GAN segmentations (0.75 (0.11) for DRIVE and 0.82 (0.091) for STARE datasets (Supplementary Fig. 4)). These values compare favourably against previous methods using manually labelled data[58] for which state-of-the-art was 0.8378 for STARE and 0.8275 for DRIVE, particularly given our absence of manual labelling and limited quantity of training data.

These results call into question how appropriate manual segmentation is as a gold standard in this setting; visual inspection suggests these additional small vessels labelled in the OCT-A data are

indeed physiological and were simply missed by manual segmentation. PI-GAN achieved high accuracy comparable with state of the art, and demonstrated potential in accessing smaller vessels in higher resolution images.

They also demonstrate the key ability of physics-informed simulations with deep learning to autonomously segment blood vessels within a range of ophthalmology imaging modalities, without the need for any manually labelled training data.

## Discussion

We have presented a physics-informed, generative approach that combines biophysical simulation with deep generative learning, to support quantitative assessment of retinal vasculature in clinical ophthalmology, alongside supporting research into the retinal presentation of systemic diseases such as cardiovascular disease and vascular dementia[3,4]. A useful outcome of this approach is the ability to automatically segment vascular data from clinical evaluation images, without any need for manual segmentation. Specifically, we created a

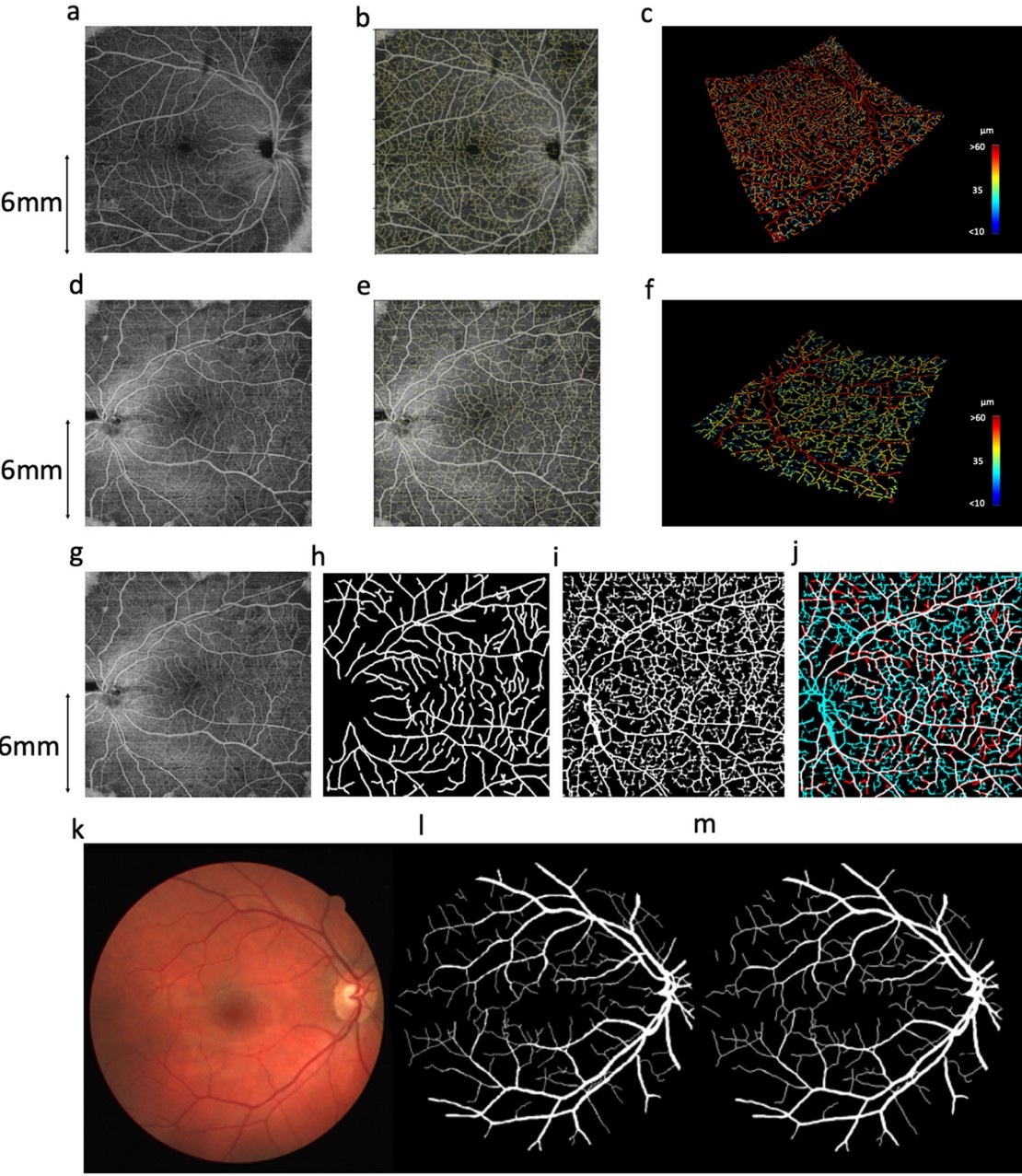

**Fig. 7 | Blood vessel segmentation from OCT-A data with PI-GAN. a–d** OCT-A en-face images of retinal vasculature. **b–e** The same OCT-A images with vessel segmentations from PI-GAN. **c–f** Segmented vessels projected in three-dimensional space, colour-coded for vessel radius. **g** An OCT-A image with (**h**) manually-segmented and (**i**) PI-GAN-segmented blood vessels. **j** A composite image of manually- and PI-GAN-segmentations, with overlapping pixels rendered white, pixels with only PI-GAN-detected vessels in blue and pixels with only manual-detected vessels in red. **k** A retinal photograph taken from the DRIVE data set[72] with (**l**) manually-segmented and (**m**) PI-GAN-segmented blood vessels. Scale bars are provided in millimeters (mm).

linked set of algorithms that draw on established principles in biophysics to simulate fully-connected retinal vasculature, in a three-dimensional domain, with special treatment for optic disc and macula regions. The full connectivity of our models, with separate arterial and venous trees, enables realistic blood flow and delivery simulations (for example, as we show in FA). We demonstrated that our synthetic vascular networks are highly concordant with real retinal vasculature metrics, with network statistics matching those from manual segmentations, in three regions: the optic disc, macula and periphery.

This close accordance between simulation and real-world geometries is key to its ability to segment blood vessels from ophthalmology images. Cycle-consistent deep generative learning allowed us to create realistic fundus photograph, OCT-A and FA images that inherently maintained feature geometry through the translation from simulation to clinical image domains. The resultant data are inherently paired, and so could provide data to augment conventional supervised learning approaches. However, cycle-consistency also facilitates the reverse translation, from clinical image domains back into the simulation domain, allowing the automated segmentation of blood vessels without human-labelled data. Comparing segmentation performance against manual segmentations revealed a much greater ability to label small vessels in high resolution images, and with excellent overlap with larger manually-segmented vessels. Overall performance assessed via Dice score for OCT-A showed a relatively low accordance, due in part to the greater ability of the PI-GAN approach to detect small blood vessels omitted from manual segmentations, but also false-positives in

human-labels, alongside a small number (0.67%) of false-positive PI-GAN-labelled vessels ('hallucinations'[59]). Benchmarking PI-GAN segmentations against publicly-available retinal photograph datasets returned Dice scores aligned with state of the art, which is remarkable for a technique that requires no human oversight.

To date, supervised deep learning approaches have yielded impressive results in 2D vessel segmentation relative to manual segmentation, although tend to favour precision over recall[5], resulting in an under-segmentation of faint vessels, underestimation of the width of thicker vessels and some 'elusive' vessels being missed[57]. This is problematic for diagnostic interpretation, because many biomarkers (such as artery-vein (AV) ratio, branching angles, number of bifurcations, fractal dimension and tortuosity) need precise measurements of individual vessels. GANs incorporating cycle-consistency have previously been used for medical imaging domain machine learning tasks such as chest MRI to X-ray CT transformation[28], PET image denoising[29], and artefact reduction in fundus photography[30]. Likewise Menten et al used the space colonisation algorithm to generate macular blood vessel images, which they coupled with deep learning[31].

Our approach builds on this by incorporating biophysically-informed models of flow within fully-connected artery and venous networks that extend across the entire retina, and our use of it to inform cycle-consistent deep generative learning. These developments allow application in larger field of view images (e.g. wide-field fundus photography), and also enable a large range of future applications, including flow modelling and oxygen delivery[60]. Moreover, given our ability to model arterial and venous trees, there is potential for independent segmentation of both vascular supplies.

These biophysical simulations also aimed to capture the wide range of variation found in real retinal networks, by varying the 26 simulation parameters across their reported physiological range. A further advantage of developing flow models into our biophysical framework was the ability to simulate pathology, such as the progression of DR and RVO. Many other pathologies could be simulated in follow-on studies, including changes in retinal vessel diameters associated with factors such as aging or hypertension. For example, Wong and colleagues reported retinal arteriolar diameters to decrease by -2.1 μm for each decade increase in age, and by 4.4 μm for each 10 mmHg increase in arterial blood pressure[61]. Performing disease-specific deep generative learning runs will enable us to further refine our segmentation approaches and begin to characterise pathology.

Accordingly, there is also potential to use clinical data to further improve our biophysical simulations, enabling more accurate modelling of retinal physiology (and disease) and the ability to develop interpretable AI systems. The results of several recent studies using deep learning suggest that that retinal vasculature can provide a window into many systemic diseases (including dementia[3], kidney disease[62] and cardiovascular disease[4]), but cannot easily explain the structural basis of these associations. A PI-GAN framework is inherently coupled to biophysical laws, and so could help determine their origins or underpinning mechanistic processes. Additional challenges for segmentation are artery-vein classification[63] and establishing connectivity of the vessels[64], which, having a well-defined ground truth data set from simulations, could be realised through PI-GAN.

Overall our results demonstrate the potential of biophysical models of the retina, which can be interrogated to understand how physiological perturbations (such as disease) effect vascular function. Further work could explore regional variability in blood flow, with the temporal side exhibiting greater flow than the nasal side in both retinal venules and arteries, which may be related to retinal ganglion cell numbers[65]. Additionally, the model could be used in predicting inhibitors of angiogenesis, such as VEGF inhibitor Bevacizumab. Incorporating this model into a larger-scale retinal model (including the choroidal supply) would enable complete simulation of the retinal supply. The ability to then apply these

simulation results for the interpretation of clinical images, via physics-informed generative learning, is a significant step forward.

## Methods

### Procedural generation of synthetic retinal vasculature

Algorithms for generating synthetic retinas followed multiple, length scale dependent steps[66]. Firstly, the values of geometrical parameters were set by sampling from a normal or uniform distribution according to parameter values shown in Supplementary Table 1. Retinal vasculature took the form of spatial graphs (i.e. branching nodes connected via vessel segments) which were initially seeded using a Lindenmayer system (L-system), and then extended using constrained constructive optimisation (CCO) and Lattice Sequence Vascularisation (LSV). For the L-system (see Github directory: retinasim/lsystemlib/) retinal artery and retinal vein segments were initially directed from a central optic disc point and branched by five generation to form seeding retinal arcades. These were passed into the CCO algorithm, followed by LSV, to further branch the vessels in progressively finer resolutions. Once complete, the macula region was cleared and regrown into a radially-spaced lattice (https://github.com/AndrewAGuy/vascular-networks). Finally, spatial undulations were imposed onto the vessels by overlaying sinusoidal curves with randomised frequencies and amplitudes (retinasim/sine_interpolate.py). These implementation of these linked algorithms is described further in the sections below.

### Lindenmeyer system seeding

Firstly, seeding networks following approximate retinal vascular branching geometry were constructed, starting with a putative central retinal artery and retinal vein positioned at the centre of the optic disc. The diameters of the retinal artery and vein were $135 \pm 15$ μm and $151 \pm 15$ μm, respectively[35], oriented parallel to the optic nerve (defined as the z-direction). Two branches were added to the end of each of these segments, oriented in the x-y plane, and one directed above and the other below the retinal midline. Subsequent branching of these vessels was performed stochastically, with segment lengths between bifurcations set as a fixed fraction of vessel diameter ($18 \pm 3$) and bifurcation vessel diameters set according to:

$$\cos \theta_1 = \frac{(1+\alpha^3)^{\frac{4}{3}} + \alpha^4 - 1}{2\alpha^2 (1+\alpha^3)^{\frac{2}{3}}} \tag{1}$$

$$\cos \theta_2 = \frac{(1+\alpha^3)^{4/3} + 1 - \alpha^4}{2(1+\alpha^3)^{2/3}} \tag{2}$$

Normally-distributed noise was added to branching angle values, with a standard deviation of 5°. Vessel bifurcation angles were assigned such that the larger vessel oriented towards the macula to create putative major vessels oriented around the macula. Fifth-order bifurcations were added to the network, or until vessels breached the edge of the retina domain. Supplementary Fig. 1a shows an example of an L-system seeding network.

### Major vessel growth

Seed vessel networks were used as input into a multi-scale growth algorithm for the creation of hierarchical vasculature. First, seed networks were amended to provide a uniform distribution of leaf nodes (terminating arteriole and venuole nodes created prior to construction of capillary networks at a later stage) throughout the circular domain, using Accelerated Constrained Constructive Optimisation[40], using a leaf node spacing of 3 mm. Multiscale, two-dimensional lattices were defined (stride lengths ranging from 3000 to 150 μm, with five

iterations linearly spaced within that range) and used to grow vessel networks by progressively adding vessels into unoccupied lattice sites from neighbouring occupied sites, choosing the candidate vessel which minimised the expected change in network cost (see below), and progressively reducing the length scale when no more progress could be made. After the initial growth stage, all existing leaf nodes were removed[67]. At all stages of the major vessel growth the macula region was kept free of vessels by removing vessels which intersected it, forcing flow to divert around it.

As retinal vasculature is positioned in front of the retina itself, we optimised networks to minimise the area of the retina occluded by vessels, according to a cost function based on Murray's law:

$$C(B, \lambda, \rho) = \sum_{b \in B} r_b^\rho l_b^\lambda \qquad (3)$$

with $\rho = 1$ and $\lambda = 1$, and where $B$ is the set of vessel segments in the network, with length $l$ and radius $r$. After each growth step the network geometry was optimised by moving vessel nodes, and highly asymmetric bifurcations were trimmed for regrowth[40] using the thresholds from ref. [39] to account for the high asymmetry of optimal networks[39]. After growth at each length scale was terminated, the networks were optimised topologically by allowing asymmetric bifurcations to move their low-flow side downstream and branches which were short compared to their expected length under the West, Brown and Enquist model[68] to be treated as a single higher-order split for regrouping using a method similar to[69]. Due to the two-dimensional nature of the networks, network self-intersections were tested for using the approach of[40] however, rather than resolving the intersections by making excursions around the contact site we rewire the vessels to prevent future iterations from recreating the same intersection.

Unlike the implementation of ref. [40], leaf nodes were allowed to move from their nominal location up to a specified "pinning distance", given as a fraction of the leaf spacing. Existing vessels could be specified as frozen, in which case the optimiser did not touch them. This approach was used to perturb the optimal root vessel structure with artificial tortuosity, strip away the downstream branches and regrow the downstream vessels, repeating this down the tree structure.

## Macula growth

Vessels supplying the macula have a characteristic radial structure, motivating the development of a particular approach to enforce this structure. This uses the same lattice site invasion approach between the macula outer radius and the fovea (which is kept vessel-free), but with the stride set low enough that the majority of the growth arises from spreading over many iterations at the same length scale rather than hierarchical refinement. The macula has a configurable flow rate density compared to the rest of the retina, ranging from 1.5 – 2.0 and leaf nodes are offset by uniformly sampling an offset in a disc around the nominal position to ensure that vessels did not align along the lattice sites. The macular vessels were prevented from doubling back on themselves by setting a hard limit on the vessel angle, preventing obviously non-physiological structures from arising whilst still allowing the radial pattern to develop. After all leaf nodes are created, a sparsity factor is specified and each leaf node removed with this probability, then the remaining vessels are geometrically optimised.

## Network overpass and interleaving

In the final stage, the arterial and venous networks have their collisions resolved using the method of ref. [40], creating out-of-plane excursions around contact sites between the networks. To enable further micro-scale network growth techniques to create an interdigitated structure, we remove the low-flow side of all arterio-venous intersections with a radius below a critical value (5 um), leaving surviving vessel geometry untouched. Interdigitations were then created using a Space Colonisation implementation[70], interspersed with geometric optimisation.

## Vessel tortuosity

The multi-scale growth algorithm creates relatively straight paths between branching points, and to simulate tortuous retinal vessels, particularly in veins, sinusoidal displacements were overlaid. Two oscillations were superimposed according to:

$$d'(x, r) = d(x, r) + a_0 \sin\left(\frac{x}{\tau_0(r)} + \delta_0\right) + a_1 \sin\left(\frac{x}{\tau_1(r)} + \delta_1\right) \qquad (4)$$

where $d(x,r)$ is the path taken by a vessel with radius $r$, and $d'(x,r)$ is the modulated path. The amplitude of displacements, $a_0 + a_1$ ranged from $r$ –3.5$r$ for arteries and $r$ –7.5$r$ for veins, with a low frequency period ($\tau_0$, ranging from 15$r$ – 25$r$) and a high frequency period ($\tau_1$, ranging from 30$r$ – 50$r$). The phase of the modulations, $\varphi_0$ and $\varphi_1$, enabled modulations to be matched between vessel bifurcations.

## Simulating vascular flow and fluorescein delivery

Blood flow in retinal networks were simulated using our REANIMATE platform[47], which uses a connectivity-based formalism to optimise Poiseuille flow in tree-like spatial graphs. As anterior retinal vasculature features a single arterial inlet and venous outlet, the system requires only one pressure boundary condition (the difference between arterial and venous inlet pressures), which was fixed at $56.2 \pm 14.0$ and $20.0 \pm 10.0$ mmHg, respectively.

Time-dependent delivery of contrast agent (e.g. fluorescein) was simulated as described in d'Esposito et al.[47]. Briefly, a bolus of fluorescein was simulated according to

$$C(t) = s_1 G_1(t; t_1, \sigma_1) + s_2 G_2(t; t_2, \sigma_2) + a_0 e^{-(t-t_3)} \qquad (5)$$

where $C(t)$ is the concentration of fluorescein as a function of time $t$. The first two terms, Gaussian functions, represent the first and second pass of the bolus and the third term, an exponential decay, represents the washout phase[51] The width of the first and second pass were $\sigma_1 = 10$ s and $\sigma_2 = 25$ s, respectively, and the decay rate of the washout phase, β, was 0.043/minute. $T_1$, $t_2$ and $\tau$ are the time to peak for the first pass, second pass and washout phases, and were set at 0.171, 0.364 and 0.482 min, respectively[51]. $S_1$, $s_2$ and α were fixed at 0.833, 0.336 and 1.064 (dimensionless units). Peak concentration was normalised to unity at the inlet to the retinal artery and the time course in each connected vessel segment was time-shifted according to the velocity of blood in each vessel and scaled according to the ratio of flow in the parent and child vessels at bifurcation points.

## Image datasets

This study was carried out in accordance with the Declaration of Helsinki[71]. Ethical approval of retrospective audit data was obtained through Moorfields Eye Hospital Research and Development and Audit ROAD17/034. Informed consent was not obtained for retrospective data. Participants were not compensated as retrospective data was utilized. Only de-identified retrospective data was used for research, without the active involvement of patients. Sex was self-reported by study participants. Clinical ophthalmological retinal images were obtained from equipment at Moorfields Eye Hospital NHS Trust, London, UK: OCT-A images were obtained from a PLEX Elite 9000 (Carl Zeiss Meditec LLC, Dublin, CA, USA), ultra-wide true colour retinal photographs were obtained from Zeiss Clarus 500 Fundus machine (Carl Zeiss Meditec LLC, Dublin, CA, USA), fluorescein angiograms were obtained from Optos widefield camera (Optos, Inc. Marlborough, MA, USA). 19 manually segmented OCT-A images were obtained from healthy controls not ascertained for disease status (10 male, 9 female, mean age 39.89 (s.d. 11.25)). These manual segmentations were used in

comparison of network structure with simulated networks. Datasets of 570 FA images, 590 colour retinal photographs, 43 OCT-A en-face images, and 130 simulated networks were used in training and testing the PI-GAN algorithm.

## Manual labelling of clinical data

Manually labelled data was generated using a custom-built Python package enabling tracing of vasculature in 3D. The process involved placing user defining control points on the 2D image indicating where in a slab the vessel is located via maximum intensity projection. The z-height of the vessel was then fixed by identifying the height of the highest signal intensity voxel, which was manually constrained to exclude the choroid or RPE. The radius of each vessel was automatically calculated by setting a user-defined signal intensity threshold. Review of segmented structures was performed in 3D panel to assess and ensure labelling quality. In images with pathological blood vessels such as DR the abnormal vasculature or areas of neoangiogenesis were traced in the same manner. Vessel information (vessel coordinates, edge connectivity, number of edge points, edge point coordinates, radii, and vessel type) was exported and stored in Amira spatial graph format (ThermoFisher Scientific, Waltham, Massachussetts USA). Retinal regions were labelled. The macula was defined as a 5.5 mm diameter circular area centred on the fovea. The vessels surrounding the optic disc were labelled as a 3.6 mm diameter centred at the optic disc. Vessels outside these regions were defined as 'peripheral'.

## Deep generative learning

Image-to-image translation was performed using cycle-consistent generative adversarial networks[30]. This algorithm enables automated unsupervised training with unpaired samples, learning a bi-directional mapping function between two different domains with deep generative adversarial networks. It utilises cycle consistency, where the reconstructed image obtained by a cycle adaptation is expected to be identical to the original image for both generative networks. Cycle-consistent GANs are composed of two main deep neural network blocks which are trained simultaneously: an image generator (generator) and an adversarial network (discriminator). There is a loss (G loss) to make a synthesised image from domain A closer to a real image from domain B, and a loss (D loss) to distinguish the synthesised image from domain A from a real image from A. There are also losses facilitating the conversion in the opposite direction (G loss making synthesised image from domain B closer to domain A, and D loss to distinguish synthesised and real domain B images). Additionally, cycle loss is the difference between the input image and the double-synthesised image and identity loss is the difference between output and input images. A train/validation/test split of 75%/5%/20% was used. All PI-GAN training and evaluation was performed using a single NVIDIA Titan RTX GPU.

We iteratively trained a switchable PI-GAN algorithm with 500 epochs (Supplemental Fig. 5). All networks were trained using the optimizer ADAM solver[35] with $\beta_1 = 0.5$, $\beta_2 = 0.999$. The learning rate for the first 100 epochs was $2 * 10^{-4}$, and then linearly decayed to $2 * 10^{-6}$. Images were pre-processed with crop size 256 pixels. The minibatch size was 1. The loss weights $\lambda$ were set as 10. The model was trained on NVIDIA TITAN RTX in Pytorch v1.9.1.

## Loss functions

**Adversarial losses.** For mapping function $G : X \rightarrow Y$ and its discriminator $D_Y$, the objective is expressed as:

$$L_{GAN}(G, D_Y, X, Y) = \mathbb{E}_{y \sim p_{data}(y)}[logD_Y(y)] + \mathbb{E}_{x \sim p_{data}(x)}[\log(1 - D_Y(G(x)))], \quad (6)$$

Where G attempts image generation G(x) to look similar to images from domain Y and $D_Y$ tries to distinguish between translated samples G(x) and real samples $y$[55].

**Cycle consistency loss.** Cycle consistency loss is utilised to reduce the space of possible mapping functions to the target domain; the image translation cycle aims to bring an image back from its translation to the original. The images should satisfy both forward and backward cycle consistency. The mapping from one image, to translated domain, back to original is incentivised using the cycle consistency loss:

$$\mathcal{L}_{cyc}(G, F) = \mathbb{E}_{x \sim p_{data}(x)}[||F(G(x)) - x||1] + \mathbb{E}_{y \sim p_{data}(y)}[||G(F(y)) - y||1] \quad (7)$$

**Identity loss.** An additional loss is utilised to regularise the generator to be near an identity mapping when real samples from the target domain are used as the input of the generator:

$$\mathcal{L}_{identity}(G, F) = \mathbb{E}_{y \sim p_{data}(y)}[||G(y) - y||1] + \mathbb{E}_{x \sim p_{data}(x)}[||F(x) - x||1] \quad (8)$$

**Statistical evaluation of synthetic vessel networks.** Vessel metrics of vessel branching angle, length, tortuosity, network volume and diameter were calculated. Two-sided Analysis of variance (ANOVA) was used to assess differences in these metrics by retina region (optic disc, macula, and periphery) and by status (healthy control and simulated network) (Supplementary Table 2) with eye (right OD/ left OS), participant sex, and OCT-A imaging scan pattern used as covariates.

**Expert evaluation of PI-GAN vessels.** A clinical expert (consultant ophthalmologist at Moorfields Eye Hospital with fellowship-level sub-specialty training in medical retinal disease and extensive clinical experience (14 years of experience)) reviewed the original segmentations from DRIVE and STARE datasets along with the corresponding images. Using evaluation in regions 'disc', 'centre' and 'mid-periphery', they evaluated all reviewed images as containing 'possible' small vessels. Their evaluation was that for 90% of DRIVE and 80.95% of STARE images, PI-GAN segmented additional vessels that were not seen in the original segmentation, that they considered 'possibly' present in the retinas. The remaining images were ranked 'unclear'. They noted that image quality for DRIVE/STARE datasets was a limiting factor in conclusively determining the presence of smaller vessels.

## Evaluation metrics

**Frechet inception distance.** GAN output was evaluated using the Fréchet Inception Distance (FID), which evaluates model quality by calculating the distance between feature vectors for real and generated images. FID compares the distribution of generated images with distribution of real images that were used to train the generator. Lower FID scores indicate more similarity between two groups. The FID score is calculated by first loading a pre-trained Inception v3 model. The output layer of the model is removed and the output is taken as the activations from the last pooling layer, a global spatial pooling layer.

Three FID scores were calculated: real simulation images (synthetic space) versus manually segmented vasculature clinical images; real retinal photographs (retinal photograph space) versus PI-GAN generated retinal photographs; real OCT-A images (OCT-A space) versus PI-GAN generated OCT-A images; real FA (FA space) versus PI-GAN generated FA images.

**Dice score.** Dice scores were additionally calculated. This is a commonly used performance statistic for evaluating the similarity of two samples. For a ground truth segmentation label L and associated

prediction P, we measure the binary Dice score D:

$$D(P,L) = \frac{2|L \cap P|}{|P| + |L|} \qquad (9)$$

We carried out benchmarking of the PI-GAN algorithm against other models trained for manual segmentations from segmentation of retinal vessels using STARE, and DRIVE datasets public datasets, which are regularly used for benchmarking of algorithm results[19,72]. Dice score were evaluated from the output of PI-GAN trained to carry out the mapping between simulated data segmentations and retinal photographs and compared to GAN performance without synthetic data.

### Reporting summary

Further information on research design is available in the Nature Portfolio Reporting Summary linked to this article.

## Data availability

All data supporting the findings are available in the article, the Supplementary Information, and from the corresponding author upon request, aside from data that is subject to the restrictions detailed below. Source retinal simulation data have been deposited on FigShare (https://figshare.com/articles/dataset/sim00000000_zip/26364181, https://doi.org/10.6084/m9.figshare.26364181) and Dropbox (https://www.dropbox.com/scl/fo/whwru5rmz8g7cr0h8ytg1/h?rlkey=ynbh2kdhe0pcvpfo6cypm9oc6&dl=0). The Moorfields Eye Hospital data is protected and subject to restrictions of data sharing due to its sensitive nature and privacy concerns, as it is patient or research participant data and is not available for access to the wider research community without data share agreements. Access to the data is managed by the audit department and Moorfields Research and Development, along with timeframes to responses to requests, and details of restrictions on data use via data use agreements (https://www.moorfields.nhs.uk/research/). DRIVE and STARE datasets are available online. Use of data from these online sources complies with database terms and conditions.

## Code availability

All code is available in https://github.com/simonwalkersamuel/retinasim/tree/main[66]. RetinaSim, and relies on several libraries: (1) The code in this repository (RetinaSim) is written in python (3.8), and both provides functionality and glues together the other libraries; (2) Reanimate for 1D flow simulation (provided here as a submodule); (3) RetinaGen for procedural modelling of blood vessel networks (provided here as a submodule); (4) Pymira for creating and editing spatial graph structures in python (provided here as a submodule). Installation instruction is provided on the GitHub repository. Deep generative learning was performed using an adaptation of https://github.com/junyanz/pytorch-CycleGAN-and-pix2pix provided in the RetinaSim codebase.

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

## Acknowledgements

This research was funded by Cancer Research UK (C44767/A29458 and C23017/A27935) and EPSRC (EP/W007096/1). Research on Anonymised Data (ROAD) approval 17/034 was used in accessing ophthalmological image data from Moorfields Eye Hospital NHS Foundation Trust. The audit was authorised by Moorfields Clinical Audit team. EB is a part of EPSRC-funded UCL Centre for Doctoral Training in Intelligent, Integrated Imaging in Healthcare (i4health). CW was supported by the Chan Zuckerberg Initiative DAF an advised fund of Silicon Valley Community Foundation (2020-225394). The authors would like to thank Drs Gabriela Grimaldi and Rajna Rasheed for assigning artery-vein status to a subset of retinal photographs (n=5), and Henry Cole, Andrew. Kume, Jinyu Li, Kendra Hilliard, Yiyun Zhang, Jiahao Xu, Shuo Wu who assisted with data pre-processing, labelling and segmentation. We also thank all participants and patients who contributed ophthalmological imaging data.

## Author contributions

E.E.B. contributed conception and design, analysis and interpretation of data, creation of software, and drafting the work. A.A.G. contributed creation of new software used. N.A.H. contributed analysis and interpretation of data. P.S. contributed creation of new software used. L.G. contributed analysis and interpretation of data. H.C. contributed analysis and interpretation of data. C.W. contributed analysis and interpretation of data. AM contributed creation of new software used. R.S. contributed conception and design, acquisition, and substantial revision to draft. R.R. contributed conception and design of work, acquisition of data, interpretation of data, and substantially revising draft. S.W.S. contributed conception and design of work, acquisition, interpretation of data, creation of new software, and substantial revision of draft.

## Competing interests

The authors declare no competing interests.
