## [Peer Review File · Nature Communications]

Physics-informed deep generative learning for quantitative assessment of the retinaREVIEWER COMMENTS

Reviewer #1 (Remarks to the Author):

This is an interesting paper from a truly multidisciplinary team of investigators with a strong track record. The study addresses the challenge of how to extract the retinal vasculature from clinical images with the potential to use the data to inform physiological models. Retinal vein occlusion and diabetic retinopathy are given as instances of pathological processes that can be explored using the described tools.

Novel contributions include parametrised, simulated models of retinal circulation and AI – informed segmentation of clinical images (fundus photographs and OCTA images) with capacity for inverse transformation from synthetic reconstructions to clinical image representations.

The generation of physics-inspired vascular networks that are informed by clinical images is an important advance. The models generated, as well as having intrinsic value, provide a framework from which more advanced, sophisticated models can be developed.

The rules for the creation of the pathological models are quite simple but the resulting images are perhaps surprisingly close to those seen clinically. This of course doesn't mean that the rules are 'correct' but the framework offers great potential for exploration of pathological processes.

A few minor points

Figure 3: Legend should include details of the meaning of the different elements in the figure

Figure 4: I am not sure how clearly these figures will reproduce. I had to turn the brightness up very high on my laptop to see the detail! Similar comments apply to some of the other figures. A higher power image of a reconstructed capillary bed might be of interest.

I understand that 'fake' has become a generally used adjective for synthetic images. It still feels a bit uncomfortable to me! Is the term 'synthetic' less emotive?

I get the maths provided intuitively but don't feel equipped to confirm its accuracy.

A tiny point. At places such as L512, where used as an adjective and not a noun, macula should be macular.

In Discussion:

It might help the reader to explain that the model doesn't get generate the fine-detail of the different capillary plexi at different levels within the retina.

In Supplementary material:

Parameter table: the figure for eye radius is actually eye diameter

Supplementary figure 1A and B, 3 A-C: there is a horizontal pale strip. Is this part of the model or artefact of plotting?

Software:

I note some of the links to the code are not active yet but that code is accessible from the pre-print. I haven't yet had the opportunity to run the code myself.

Reviewer #2 (Remarks to the Author):

Paper summary:

This paper proposes a physics-informed retinal vascular segmentation synthesis, including blood flow dynamics and contrast injection that behaves like real data. From these segmentations, a Deep Learning Cycle-GAN method is employed to successfully produce three different modalities of retina images. The authors apply these segmentations-to-image method in reverse to also produce segmentations from real images.

The method for generating vessel segmentation is very thorough: the authors simulate retina vasculature by considering the physics governing it, drawing a lot of clinical data to make the simulation as realistic as possible. They then generate fundus, OCT-A and FA

images with high visual quality, including pathological scenarios using three Cycle-GAN networks. They report blood flow and other clinical metrics derived from the images that are consistent with that of real data. For the segmentation task, which employs the Cycle-GANs in reverse (from images to segmentations), the achieved Dice score is very low, although sensitivity is high.

Major comments:

-The authors do not mention any similar methods for segmentation and image synthesis, although some works are available in the literature (such as Andreini et al. – 2022). If not incorporating these methods as a baseline, the authors should at least reference these methods in their introduction.

-The segmentation experiment design appears to be flawed, due to the fact that manual segmentations are coarser than the simulated vasculature. When training a Cycle-GAN, the segmentations it sees are detailed, which explains the difference between the segmentations obtained with the reverse cycle process and the ground truth manual labels. These segmentations work fine for the experiment depicted in Figure 3, but not for the Dice. Authors should be clearer about this and use a comparable baseline if it exists (i.e. a trained U-net or U-net-like network that provides finer segmentations). This problem would not explain, however, false positives hallucinated by the GAN and not present on the original simulated segmentation. Authors could forward a simulated vasculature image to obtain an image, and then obtain the segmentation by reverse process, and then calculate the Dice; giving an idea of how much is being hallucinated.

-Authors could train the Cycle-GAN on Dice (for the image to segmentation branch) loss, instead of using MSE loss, as it makes more sense since they are working with sparse, categorical data.

-Following up from the previous comment, the authors mention that the original segmentations are missing “small” vessels. This should be backed up at least by a qualified/experienced rater opinion, whose experience should be reflected on the paper.

-Something similar happens with the image quality: FID is the metric the authors use, and they report “low FID values”. Nonetheless, FID is a boundless metric, and hard to interpret without being compared to other methods (i.e. Andreini et al.’s). Another option would be calculating the FID between two sets of real images and see how it compares to the one

achieved between real and synthetic.

-The methodological section was hard to understand without a diagram. Figure 1 does not make clear enough what is the flow: elements representing the software/method should be depicted between output types (see example figure in attached PDF). Likewise, figure with the GAN architecture would be useful (depicting the different losses).

-In Figure 3, could the authors explain why there are so many outliers in the distributions? Maybe show examples of what these outliers correspond to? There are so many.

-Simulation of vasculature: In addition, the sections like “Major vessel growth” are also confusing; another logic diagram depicting the decisions and parts of the process would help (i.e., initial growth stage, removal of leaf nodes, moving vessel nodes).

-the Github link does not work. Even though it is referred, the authors should briefly mention what type of platform it is and how it works. When the authors say “networks in the form of spatial graphs were constructed using multiple linked algorithms” (L. 438-440) what do they mean?

Minor comments:

- The first paragraph of the discussion should go more in the Introduction, as it is a motivation for the work (L. 308 – 319).

- The concept of FA is brought about in the “Results” section (L. 182). It should be advanced in the introduction for further understanding (around line 59).

-In the Cycle-GAN section (Deep Generative Learning), in line 603, 604 etc., the authors keep referring to domains A, B, C and D. Consider defining these, or use a more intuitive name (OCT-A space, FA space etc.). Equations for the losses should also be provided.

-Figure 5: instead of comparing a segmentation mask with a real image, either simulate the image corresponding to that mask and display 1. Synthetic segmentation 2. Synthetic image 3. Real image, or display the real segmentation. It’s hard to visually compare like this. For the DR sub-figures, arrows should point to relevant areas in the image presenting this occlusion.

Other comments and typos:

-L. 189 – “We simulated the systemic pharmacokinetics of fluorescein using literature data” (this takes you to a graph, but what is the exact data this came from?)

-Diabetic retinopathy (DR) should be first mentioned along with its acronym (in the abstract and/or the main body) (L. 15)

-FID needs citation

-DICE should be Dice (L. 279)

REVIEWER #1

Comments to the Author

This is an interesting paper from a truly multidisciplinary team of investigators with a strong track record. The study addresses the challenge of how to extract the retinal vasculature from clinical images with the potential to use the data to inform physiological models. Retinal vein occlusion and diabetic retinopathy are given as instances of pathological processes that can be explored using the described tools.

Novel contributions include parametised, simulated models of retinal circulation and AI – informed segmentation of clinical images (fundus photographs and OCTA images) with capacity for inverse transformation from synthetic reconstructions to clinical image representations. The generation of physics-inspired vascular networks that are informed by clinical images is an important advance. The models generated, as well as having intrinsic value, provide a framework from which more advanced, sophisticated models can be developed.

The rules for the creation of the pathological models are quite simple but the resulting images are perhaps surprisingly close to those seen clinically. This of course doesn't mean that the rules are 'correct' but the framework offers great potential for exploration of pathological processes.

We acknowledge the dedication of the reviewer in reviewing the article, for their positive feedback and their educative clues for improving manuscript and figure readability. We have followed all the points raised by Reviewer #1 as detailed below.

Minor comments

1. Figure 3: Legend should include details of the meaning of the different elements in the figure

Response 1: We agree with Reviewer #1 that more details would provide clarity and have amended Figure 3 legend (L194-196) to include a description of all listed elements: branching angle, vessel length, vessel tortuosity, vessel volume, vessel diameter. It had previously included description of healthy control, periphery, macula, and optic disc elements.

2. Figure 4: I am not sure how clearly these figures will reproduce. I had to turn the brightness up very high on my laptop to see the detail! Similar comments apply to some of the other figures. A higher power image of a reconstructed capillary bed might be of interest.

Response 2: The resolution of the original image is sufficient to see the detail of the capillary bed as can be seen in the attached PowerPoints showing the capillary beds (Figure 2 and Figure 5). We are therefore able to provide a higher resolution image but the image resolution is limited by the file size restrictions in place. The higher resolution image would allow the figure to reproduce.

3. I understand that 'fake' has become a generally used adjective for synthetic images. It still feels a bit uncomfortable to me! Is the term 'synthetic' less emotive?

Response 3: We concur that 'synthetic' more appropriately describes the images than 'fake'. We amend all four uses of the word 'fake' to 'synthetic' to describe images in the main body of the text at L300, L395, and L398. We additionally amended Figure 1 (L107) by removing the word 'fake' and replacing this with 'synthetic'.

4. I get the maths provided intuitively but don't feel equipped to confirm its accuracy.

Response 4: We provide a further overview of the multiple linked algorithms at L612-616. We have additionally included all relevant equations in the methods at L628-633, L658-660, L706-709, L725-728, L798-800, L811-814, L818-820, and L878-879.

5. A tiny point. At places such as L512, where used as an adjective and not a noun, macula should be macular.

Response 5: We reviewed all instances of 'macula'/'macular' use and have amended these at positions L110 ('macula' as noun), L446 ('macula' as noun), and L688 ('macular' as adjective for vessels). These occurred at locations in the text where macula/r could be plausibly interchanged as noun or adjective, i.e. 'macula/r and optic disc features'; we amended to ensure consistency.

6. In Discussion:

It might help the reader to explain that the model doesn't get generate the fine-detail of the different capillary plexi at different levels within the retina.

Response 6: We acknowledge the reviewer in making this important point regarding how capillary plexi in our model correspond to those in real human retinas. The algorithm we generated and described in the manuscript creates retinal vasculature corresponding to established research [4, 5- supplemental manuscript citations] (Supplemental Table 1), including capillary plexi. However, insufficient data is available regarding capillary plexi in different levels of the retina and how they interlink; this is an area of active research and there are challenges in elucidating these structures due to artefacts in in-vivo imaging such as projection artefact [Campbell, J.P., Zhang, M., Hwang, T.S., Bailey, S.T., Wilson, D.J., Jia, Y. and Huang, D., 2017]. Detailed vascular anatomy of the human retina by projection-resolved optical coherence tomography angiography. Scientific reports, 7(1), p.42201], as well as any distortion that may occur when imaging post-mortem tissue. Other studies were undertaken in non-human primates. The synthetic model of the retina incorporates a 3D instantiation of retinal capillary plexi and the resolution of smaller vessels is high, as indicated by the analyses described in the main body of the review and in the results section.

7. In Supplementary material:

Parameter table: the figure for eye radius is actually eye diameter

Response 7: We have amended 'radius' to 'diameter' (Supplementary material, parameter table, L2)

8. Supplementary figure 1A and B, 3 A-C: there is a horizontal pale strip. Is this part of the model or artefact of plotting?

Response 8: The synthetic retinas feature a strict demarcation of the midline which could be improved in a future iteration; notably, as shown in the statistical analysis (Supplemental table 2), the synthetic vasculature was not statistically significantly different from real retinal vasculature in metrics of branching angle, vessel length, vessel tortuosity, vessel volume and vessel diameter.

9. Software:

I note some of the links to the code are not active yet but that code is accessible from the pre-print. I haven't yet had the opportunity to run the code myself.

Response 9: The code base is active via the link provided here: <https://github.com/simonwalkersamuel/retinasim> and in the section 'Code availability' (L584). This was previously provided via the Code Checklist submission item.

Comments to the Author

This paper proposes a physics-informed retinal vascular segmentation synthesis, including blood flow dynamics and contrast injection that behaves like real data. From these segmentations, a Deep Learning Cycle-GAN method is employed to successfully produce three different modalities of retina images. The authors apply these segmentations-to-image method in reverse to also produce segmentations from real images.

The method for generating vessel segmentation is very thorough: the authors simulate retina vasculature by considering the physics governing it, drawing a lot of clinical data to make the simulation as realistic as possible. They then generate fundus, OCT-A and FA images with high visual quality, including pathological scenarios using three Cycle-GAN networks. They report blood flow and other clinical metrics derived from the images that are consistent with that of real data. For the segmentation task, which employs the Cycle-GANs in reverse (from images to segmentations), the achieved Dice score is very low, although sensitivity is high.

We acknowledge the detailed comments from Reviewer #2 and have implemented all suggestions regarding the analysis as indicated below.

Major comments:

1. The authors do not mention any similar methods for segmentation and image synthesis, although some works are available in the literature (such as Andreini et al. – 2022). If not incorporating these methods as a baseline, the authors should at least reference these methods in their introduction.

Response 1: We have taken the paper referred to by Reviewer #2 to be ‘Andreini, P., Ciano, G., Bonechi, S., Graziani, C., Lachi, V., Mecocci, A., Sodi, A., Scarselli, F. and Bianchini, M., 2021. A two-stage gan for high-resolution retinal image generation and segmentation. Electronics, 11(1), p.60.’, as we did not have access to the exact citation and this paper is relevant. We added a sentence clarifying general previous approaches to L43-46: “Novel deep learning architectures and hyperparameter selection are often employed to mitigate issues of smaller or lower quality datasets [13, 14], though there is consensus that the input data is of primary importance.” Here we referenced work on attention, graph, and semi-supervised approaches. We also added a statement later in the paragraph (L52-57) referencing the Andreini paper referred to by Reviewer #2 and other GAN based and synthetic approaches to both retinal and other blood vessel segmentation: “Previous studies have successfully used GAN based segmentation to segment retinal fundus [21] and OCT-A [22] images, without the need to create a generalisable model. Synthetic data approaches have also been deployed for blood vasculature on experimental imaging datasets (mesoscopic photoacoustic imaging) [23]. Approaches that utilise and extend these techniques for application on high resolution wide-field images are urgently needed to enable the robust translation of deep learning into the clinic.”

2. The segmentation experiment design appears to be flawed, due to the fact that manual segmentations are coarser than the simulated vasculature. When training a Cycle-GAN, the segmentations it sees are detailed, which explains the difference between the segmentations obtained with the reverse cycle process and the ground truth manual labels. These segmentations work fine for the experiment depicted in Figure 3, but not for the Dice. Authors should be clearer about this and use a comparable baseline if it exists (i.e. a trained U-net or U-net-like network that provides finer segmentations). This problem would not explain, however, false positives hallucinated by the GAN and not present on the original simulated segmentation. Authors could forward a simulated vasculature image to obtain an image, and then obtain the segmentation by reverse process, and then calculate the Dice; giving an idea of how much is being hallucinated

Response 2: We acknowledge reviewer #2 raising this crucial difference between the finer detailed segmentations from the simulated networks and the manual segmentations from public datasets STARE and DRIVE. While we the authors agree with reviewer #2 that the reason the PI-GAN generates more detailed segmentations is due to the use of the finer simulated vasculature in training, our existing argument in the paper is that the use of simulated data in PI-GAN training improves the previous ‘ground truth’ segmentation (DRIVE/STARE) through extraction of more detailed vasculature.

We argue that this is facilitated by the simulated data featuring smaller vessels which enable labelling of finer vessels by the PI-GAN algorithm, including vessels that may be not be as apparent to a human observer, for instance rated by human observers as ‘possible’ or ‘unclear’. As described in the below response 4 to Reviewer #2, the resolution of these datasets is low which causes challenges in expert vessel identification. We note that the Dice score is relatively low, which we argue is due to the differences in segmentation of smaller vessels. We acknowledge Reviewer #2’s comments on further clarifying the nature of smaller vessels segmented using PI-GAN. As the images are lower resolution meaning PI-GAN labelled smaller vessels were difficult to validate conclusively, we have undertaken a substantial revision.

Reviewer #2 commented that we should use a comparable baseline because the simulated datasets contained smaller vessels than the original manual segmentations. Therefore, simulated vessel data without capillary-level detail was also generated and PI-GAN was again trained. This analysis achieved a higher DICE score compared to original synthetic vasculature Dice scores (calculated for PI-GAN segmentation and manual segmentation) for DRIVE (mean 0.75 (s.d. 0.11) reduced vasculature compared to mean 0.56 (s.d. 0.013) original vasculature) and STARE (mean 0.82 (s.d. 0.091) compared to mean 0.64 (s.d. 0.19)) (Rebuttal Table 1 and Rebuttal Figure 1 below). The finding that a high Dice score is achieved on a reduced vasculature segmentation validates the theory that PI-GAN is capturing the same vessels identified through manual segmentation while also capturing additional smaller vessels. We have amended the results section to describe the finding of reduced vasculature PI-GAN extracting DRIVE/STARE vessels with high Dice score (L306-327, Figure 7, and Supplemental Figure 5).

	DRIVE dataset	STARE dataset
Mean Dice score between PI-GAN and manual segmentation (sd) (2.d.p)	0.56 (0.013)	0.64 (0.19)
Mean Dice score between Reduced vasculature PI-GAN and manual segmentation (sd) (2.d.p)	0.75 (0.11)	0.82 (0.091)

Rebuttal Table 1. Dice scores calculated for DRIVE and STARE datasets between PI-GAN and manual segmentations for original synthetic vasculature and reduced synthetic vasculature (without capillary level detail)

Rebuttal Figure 2. STARE and DRIVE images a and d) original image; b and e) original dataset manual segmentation; c and f) reduced vasculature PI-GAN generated images

3. Authors could train the Cycle-GAN on Dice (for the image to segmentation branch) loss, instead of using MSE loss, as it makes more sense since they are working with sparse, categorical data.

Response 3: We note that the current approach using MSE loss has proven effective in producing results achieving high dice scores that are competitive with state-of-the-art approaches [in manuscript citation, 58]. However, this alternative training method of using Dice for the image to segmentation branch loss could be used in a future iteration of PI-GAN.

- Following up from the previous comment, the authors mention that the original segmentations are missing “small” vessels. This should be backed up at least by a qualified/experienced rater opinion, whose experience should be reflected on the paper.

Response 4: We have added the following evaluation from a clinical expert to the methods (L847-856):

‘A clinical expert (consultant ophthalmologist at Moorfields Eye Hospital with fellowship-level subspecialty training in medical retinal disease and extensive clinical experience (28 years of experience)) reviewed the original segmentations from DRIVE and STARE datasets along with the corresponding images. Using evaluation in regions ‘disc’, ‘centre’ and ‘mid-periphery’, they evaluated all reviewed images as containing ‘possible’ small vessels. Their evaluation was that for 90% of DRIVE and 80.95% of STARE images, PI-GAN segmented additional vessels that were not seen in the original segmentation, that they considered ‘possibly’ present in the retinas. The remaining images were ranked ‘unclear’. They noted that image quality for DRIVE/STARE datasets was a limiting factor in conclusively determining the presence of smaller vessels.’

We additionally note that the PI-GAN algorithm has additionally been trained and evaluated on ultra-wide true color retinal photographs obtained from Zeiss Clarus 500 Fundus machine (Carl Zeiss Meditec LLC, Dublin, CA, USA). This machine produces a wider field of view (FOV) (133°) and higher resolution (7 μm) image. STARE database images were captured on TopCon TRV-50 fundus camera with 35° FOV and 605 x 700 pixel image, 24 bits per pixel. DRIVE database images were acquired on Canon CR5 non-mydratiac 3CCD camera with 45° FOV and 584*565 pixels with eight bits per color channel (3 channels). Though the lower resolution images form the widely used benchmark.

The issues with the public dataset in terms of missing vessels and poor quality have also been noted in the literature previously: for instance in Jin, K., Huang, X., Zhou, J., Li, Y., Yan, Y., Sun, Y., Zhang, Q., Wang, Y. and Ye, J., 2022. Fives: A fundus image dataset for artificial Intelligence based vessel segmentation. *Scientific Data*, 9(1), p.475. This citation was added to L43.

- Something similar happens with the image quality: FID is the metric the authors use, and they report “low FID values”. Nonetheless, FID is a boundless metric, and hard to interpret without being compared to other methods (i.e. Andreini et al.’s). Another option would be calculating the FID between two sets of real images and see how it compares to the one achieved between real and synthetic.

Response 5: As suggested by reviewer 2 we have implemented the Frechet Inception Distance Score (FID) between arbitrarily split sets of real images. We assigned the two groups of real images for each image type using R runif() function. We calculate FID between 1) two sets of retinal photographs, 2) two sets of OCT-A, 3) two sets of FA, as shown in the table below, which also features the previous FID results:

Imaging modality	Real versus synthetic FID	Real versus real FID
Retinal photographs	6.95	8.30
OCT-A enface	3.06	4.41
FA	5.17	3.69
Mean (sd)	6.95 (1.95)	8.30 (2.48)

Rebuttal Table 2: FID scores for real versus synthetic data and real versus real data

As seen in the above table, the FID metric indicates there is similar variability in the real versus real and real versus synthetic images for different imaging modalities. This indicates that the synthetic images demonstrated accordance with real images, to a degree that also reflects the variability between real images from the same imaging modality which feature variation in brightness, contrast, orientation, pathological features/ disease states, and image artefacts.

6. The methodological section was hard to understand without a diagram. Figure 1 does not make clear enough what is the flow: elements representing the software/method should be depicted between output types (see example figure in attached PDF). Likewise, figure with the GAN architecture would be useful (depicting the different losses).

Response 6: The authors have not received the attached PDF but have amended Figure 1 to make the flow clearer using arrows and additional text to clarify the flow of data and output types.

7. In Figure 3, could the authors explain why there are so many outliers in the distributions? Maybe show examples of what these outliers correspond to? There are so many.

Response 7: Upon reviewing the data, the main outliers are in vessel length, vessel tortuosity, and vessel diameter. The outliers in vessel diameter correspond to larger vessels near the optic disc. The vessel length outliers correspond to major vessels that can be traced across longer distances. While vessel tortuosity outliers occur due to tortuosity variation. Grubbs test performed in 'outlier' R package did not identify a statistically significant outlier for each metric and therefore there was not considered to be justification to remove outliers in boxplots from statistical analysis.

8. Simulation of vasculature: In addition, the sections like "Major vessel growth" are also confusing; another logic diagram depicting the decisions and parts of the process would help (i.e., initial growth stage, removal of leaf nodes, moving vessel nodes).

Response 8: Figure 1 (L107) has been amended to show the overall flow between elements and this has been clarified further at L612-617.

9. The Github link does not work. Even though it is referred, the authors should briefly mention what type of platform it is and how it works. When the authors say "networks in the form of spatial graphs were constructed using multiple linked algorithms" (L. 438-440) what do they mean?

Response 9: The code base is active via the link provided here: <https://github.com/simonwalkersamuel/retinasim> and in the section 'Code availability' (L585). This was previously provided via the Code Checklist submission item.

In response to Reviewer #2 comments we have added a brief description of the code functionality and operation in the 'Code availability section' (L595-L593): "All code is available in <https://github.com/simonwalkersamuel/retinasim/tree/main>. RetinaSim, and relies on several libraries: 1) The code in this repository (RetinaSim) is written in python (3.8), and both provides functionality and glues together the other libraries; 2) Reanimate for 1D flow simulation (provided here as a submodule); 3) RetinaGen for procedural modelling of blood vessel networks (provided here as a submodule); 4) Pymira for creating and editing spatial graph structures in python (provided here as a submodule). Installation instruction is provided on the GitHub repository. Deep generative learning was performed using an adaptation of <https://github.com/junyanz/pytorch-CycleGAN-and-pix2pix> described below."

As suggested by Reviewer #2, we amended the text (L612-617) to include a brief overview of the multiple linked algorithms which are then described in further detail in the subsequent method subsections: "There are four linked algorithms, the implementation of which is described in further detail in the sections below: 1) Lindemayer system (L-system) seeding, (Github directory: retinasim/lssystemlib/) 2) Initial accelerated CCO step (described in the section entitled 'Major vessel growth'), 3) Lattice sequence vascularisation with rebalancing (<https://github.com/AndrewAGuy/vascular-networks>), 4) a method of inserting perturbations at 2 frequencies (retinasim/sine_interpolate.py)."

Minor comments:

1. The first paragraph of the discussion should go more in the Introduction, as it is a motivation for the work (L. 308 – 319).

Response 1: We have incorporated the first paragraph of the discussion into the introduction (L46-L52), while leaving a sentence fragment in the discussion section L439-441 to provide a brief contextualization of the discussion.

2. The concept of FA is brought about in the “Results” section (L. 182). It should be advanced in the introduction for further understanding (around line 59).

Response 2: In response to Reviewer #2 comment we have added an introductory statement regarding the relevance of Fluorescein Angiography at L71-75, “The simulation of fluorescein angiography (FA) data, in which the time- and flow-dependent delivery of a contrast agent enables the characterisation of retinal vessels, is of particular interest as it facilitates the validation of temporal dynamics. The dye is also poorly tolerated by patients which reduces its usage clinically; simulated data could therefore be a valuable alternative [25, 26].” We additionally amended subsequent mentions of ‘Fluorescein angiography’ to ‘FA’ at L115, L200 and L206.

3. In the Cycle-GAN section (Deep Generative Learning), in line 603, 604 etc., the authors keep referring to domains A, B, C and D. Consider defining these, or use a more intuitive name (OCT-A space, FA space etc.). Equations for the losses should also be provided.

Response 3: We altered domains A, B, C, D to synthetic space, retinal photograph space, OCT-A space, FA space at all relevant positions (L294-300, L394-398, L869-872), in order to provide more intuitive understanding for the reader. The equations for the losses have been added (L794-820).

4. Figure 5: instead of comparing a segmentation mask with a real image, either simulate the image corresponding to that mask and display 1. Synthetic segmentation 2. Synthetic image 3. Real image, or display the real segmentation. It’s hard to visually compare like this. For the DR sub-figures, arrows should point to relevant areas in the image presenting this occlusion.

Response 4: This Figure is intended to display RVO/DR images that look similar to the clinical example to demonstrate disease modelling capability of the retinal vasculature generation algorithm. The figure is not intended to show correspondence between retinal vasculature in synthetic and real examples, as described by Reviewer #2.

Other comments and typos:

1. L. 189 – “We simulated the systemic pharmacokinetics of fluorescein using literature data” (this takes you to a graph, but what is the exact data this came from?)

Response 1: The parameters of the model and shape of the curve are from ‘Parker, G.J., et al., *Experimentally-derived functional form for a population-averaged high-temporal-resolution arterial input function for dynamic contrast-enhanced MRI*. Magn Reson Med, 2006. **56**(5): p. 993-1000. (citation 51)), which because no data is available for Fluorescein specifically, instead models pharmacokinetics of dynamic contrast-enhanced MRI. While the fluorescein extraction rate/ elimination rate is given by McClintic, B.R., McClintic, J.I., Bisognano, J.D. and Block, R.C., 2010. The relationship between retinal microvascular abnormalities and coronary heart disease: a review. *The American journal of medicine*, **123**(4), pp.374-e1 (citation 4). This is described further at L716-738. The line referred to by Reviewer #2 (now L224 and L227) has been amended to cite the literature data as being used for modelling fluorescein pharmacokinetics.

2. Diabetic retinopathy (DR) should be first mentioned along with its acronym (in the abstract and/or the main body) (L. 15)

Response 2: We note that diabetic retinopathy is mentioned accompanied by the acronym in the first sentence of the introduction (L28). In accordance with Nature formatting guides to avoid acronym usage in the abstract, we did not introduce the acronym when diabetic retinopathy is mentioned in the abstract (L15) (<https://www.nature.com/nature/for-authors/formatting-guide>, <https://www.nature.com/documents/commsj-phys-style-formatting-guide-accept.pdf>).

3. FID needs citation

Response 3: Citation 56 has been added at L298:

Obukhov, A. and Krasnyanskiy, M., 2020. Quality assessment method for GAN based on modified metrics inception score and Fréchet inception distance. In *Software Engineering Perspectives in Intelligent Systems: Proceedings of 4th Computational Methods in Systems and Software 2020, Vol. 1 4* (pp. 102-114). Springer International Publishing.

4. DICE should be Dice (L. 279)

Response 4: This has been amended at all locations: L310, L322, L462, and L467.

REVIEWERS' COMMENTS

Reviewer #1 (Remarks to the Author):

I have read the revised manuscript and supplementary material as well as the authors responses to the initial set of referees' comments. It seems to me as though the points raised have been addressed. I did note a further instance of confusion over radius and diameter line 142 when the radius of the eye is given as 23 - 25 mm. This is the measurement of the diameter of the human eye.

Unlike in my previous comments, I have now had an opportunity to download the code for the vessel simulations. It is a relatively complex code and I had some difficulties but these relate to my relative inexperience in programming. I think an experienced coder would have had no problems. The code appears clearly written and the detailed description in the supplementary material as well as the messages generated by the software as it runs really help in understanding the code well enough to be able to use it as a starting point for further studies. This is a most valuable resource for the community of investigators building computational models of the eye and will probably hold relevance for those interested in other microvascular networks.

In summary I consider this an important, valuable piece of work.

Reviewer #1 (Remarks on code availability):

Please see above but, yes, I was able to install and run the simulation code using the detailed instructions provided on the GitHub repository. For some reason I had to install Armadillo separately but this probably has to do with what was loaded by default into the environment I set up. The requirements.txt file helpfully managed all the other dependencies seamlessly. I was able to reproduce figures in the paper and as mentioned above I consider this a very valuable resource for the community.

Reviewer #2 (Remarks to the Author):

This paper proposes a fundus, OCT and FA retinal image segmentation method based on applying a Cycle-GAN network to vascular segmentation maps simulated using a physics informed method. The full pipeline reveals itself to be a standalone method to generate paired retinal images with annotations that can be of use to characterize the retina and improve the performance of downstream tasks. The proposed model is novel in that (1) it simulates realistic vasculature using a physics-informed, curated method (2) it produces realistic multi-modal retinal images (contrarily to examples in the literature that tend to focus on a single modality). The authors show the potential of their method to generate healthy and pathological data that follow the distribution of real data. This comparison incorporates limitations, with all observations made by the authors backed up by cited research data.

After the first review of the paper, we acknowledge that the authors have improved upon their initial manuscript by adding relevant citations and clarifications required by the reviewers. The authors have also added new results (rebuttal tables 1, 2 and rebuttal figure 2) that put some of their results more into context. As per my opinion, the methodology is sound and clear now, after incorporation of changes.

A valid link to their code has also been provided. This codebase includes information on how to run the vasculature synthesis and curation, and has substantial implementation details (e.g. Python version, commands etc.). Additions to their methodological section suggested by the reviewers (simulation of vasculature, suggested by reviewer 2, and details on Cycle-GAN training, suggested by reviewer 1), makes the method more reproducible. The only thing that is not clear in this matter is how they adapt the Cycle-GAN codebase; the code in Github does not seem to have any information on how to run this vasculature to image training. These training and inference scripts - which should, indeed, refer to the Cycle-GAN codebase - should also be incorporated into their code, as well as steps to reproduce in their specific scenario. If these scripts are present, the authors should point to them in the README file. The overall pipeline proposed in this paper is very complex, and has multiple steps, and it will help any potential user of this codebase to have the steps to reproduce broken into steps (vasculature synthesis, image synthesis). The authors also provided

example simulated data, which is very nice.

The new experiments that the authors have run and that have been included in the response document are helpful, and we recommend that the authors incorporate them into their manuscript (even if they are included in the supplementary materials). At the moment, I haven't seen any of these new results in their modified manuscript or supplementary materials (if they are included and I have not seen them, apologies for that). In rebuttal table 2, the mean FID is not necessary; the orders of magnitude between both columns of each row are enough for comparison.

Besides these last points, I am of the opinion that the authors have addressed the reviewer's comments thoroughly.

Reviewer #2 (Remarks on code availability):

I have only skimmed through the codebase - but I have not run any of the scripts.

The authors have provided relevant links to external codebases they have used, steps to reproduce, Python versions and required packages.

They also provided link to example simulated data.

As I have pointed out, though, in my comment to the authors, I cannot see anything on the vasculature to image synthesis part. A reference to the Cycle-GAN codebase is included but I think the authors should include the exact script they used to train their Cycle-GAN model and to perform inference, even if it is an exact copy of the one in the Cycle-GAN codebase (mainly because the data loading and configuration should, at least, be different). Inclusion of these files, and modification of the README to account for them, should make this codebase a useful tool for researchers in the field. If the code is there, it should be easier to find and more obvious in the README.

The model proposed on this paper is complex and I believe a more detailed instruction on how to reproduce each of the steps would be useful.

Point-by-point responses to Review comments

REVIEWER #1

Comments to the Author

I have read the revised manuscript and supplementary material as well as the authors responses to the initial set of referees' comments. It seems to me as though the points raised have been addressed. I did note a further instance of confusion over radius and diameter line 142 when the radius of the eye is given as 23 - 25 mm. This is the measurement of the diameter of the human eye.

Unlike in my previous comments, I have now had an opportunity to download the code for the vessel simulations. It is a relatively complex code and I had some difficulties but these relate to my relative inexperience in programming. I think an experienced coder would have had no problems. The code appears clearly written and the detailed description in the supplementary material as well as the messages generated by the software as it runs really help in understanding the code well enough to be able to use it as a starting point for further studies. This is a most valuable resource for the community of investigators building computational models of the eye and will probably hold relevance for those interested in other microvascular networks.

In summary I consider this an important, valuable piece of work.

Reviewer #1 (Remarks on code availability):

Please see above but, yes, I was able to install and run the simulation code using the detailed instructions provided on the GitHub repository. For some reason I had to install Armadillo separately but this probably has to do with what was loaded by default into the environment I set up. The requirements.txt file helpfully managed all the other dependencies seamlessly. I was able to reproduce figures in the paper and as mentioned above I consider this a very valuable resource for the community.

Responses to reviewer:

1. **Reviewer comment:** I did note a further instance of confusion over radius and diameter line 142 when the radius of the eye is given as 23 - 25 mm. This is the measurement of the diameter of the human eye.

Response to reviewer: We have amended this instance to diameter which is now at L154 after editorial revisions.

REVIEWER #2

Comments to the Author

This paper proposes a fundus, OCT and FA retinal image segmentation method based on applying a Cycle-GAN network to vascular segmentation maps simulated using a physics informed method. The full pipeline reveals itself to be a standalone method to generate paired retinal images with annotations that can be of use to characterize the retina and improve the performance of downstream tasks. The proposed model is novel in that (1) it simulates realistic vasculature using a physics-informed, curated method (2) it produces realistic multi-modal retinal images (contrarily to examples in the literature that tend to focus on a single modality). The authors show the potential of their method to generate healthy and pathological data that follow the distribution of real data. This comparison incorporates limitations, with all observations made by the authors backed up by cited research data.

After the first review of the paper, we acknowledge that the authors have improved upon their initial manuscript by adding relevant citations and clarifications required by the reviewers. The authors have also added new results (rebuttal tables 1, 2 and rebuttal figure 2) that put some of their results more into context. As per my opinion, the methodology is sound and clear now, after incorporation of changes.

A valid link to their code has also been provided. This codebase includes information on how to run the vasculature synthesis and curation, and has substantial implementation details (e.g. Python version, commands etc.). Additions to their methodological section suggested by the reviewers (simulation of vasculature, suggested by reviewer 2, and details on Cycle-GAN training, suggested by reviewer 1), makes the method more reproducible. The only thing that is not clear in this matter is how they adapt the Cycle-GAN codebase; the code in Github does not seem to have any information on how to run this vasculature to image training. These training and inference scripts - which should, indeed, refer to the Cycle-GAN codebase - should also be incorporated into their code, as well as steps to reproduce in their specific scenario. If these scripts are present, the authors should point to them in the README file. The overall pipeline proposed in this paper is very complex, and has multiple steps, and it will help any potential user of this codebase to have the steps to reproduce broken into steps (vasculature synthesis, image synthesis). The authors also provided example simulated data, which is very nice.

The new experiments that the authors have run and that have been included in the response document are helpful, and we recommend that the authors incorporate them into their manuscript (even if they are included in the supplementary materials). At the moment, I haven't seen any of these new results in their modified manuscript or supplementary materials (if they are included and I have not seen them, apologies for that). In rebuttal table 2, the mean FID is not necessary; the orders of magnitude between both columns of each row are enough for comparison.

Besides these last points, I am of the opinion that the authors have addressed the reviewer's comments thoroughly.

Reviewer #2 (Remarks on code availability):

I have only skimmed through the codebase - but I have not run any of the scripts.

The authors have provided relevant links to external codebases they have used, steps to reproduce, Python versions and required packages.

They also provided link to example simulated data.

As I have pointed out, though, in my comment to the authors, I cannot see anything on the vasculature to image synthesis part. A reference to the Cycle-GAN codebase is included but I think the authors should include the exact script they used to train their Cycle-GAN model and to perform inference, even if it is an exact copy of the one in the Cycle-GAN codebase (mainly because the data loading and configuration should, at least, be different). Inclusion of these files, and modification of the README to account for them, should make this codebase a useful tool for researchers in the field. If the code is there, it should be easier to find and more obvious in the README.

The model proposed on this paper is complex and I believe a more detailed instruction on how to reproduce each of the steps would be useful.

Responses to reviewer:

1. **Reviewer comment:** The only thing that is not clear in this matter is how they adapt the Cycle-GAN codebase; the code in Github does not seem to have any information on how to run this vasculature to image training. These training and inference scripts - which should, indeed, refer to the Cycle-GAN codebase - should also be incorporated into their code, as well as steps to reproduce in their specific scenario. If these scripts are present, the authors should point to them in the README file.

Response to reviewer: We have added the CycleGAN codebase used as a submodule along with instructions in the README file to the GitHub repository, in response to the comment from reviewer 2.

2. **Reviewer comment:** The overall pipeline proposed in this paper is very complex, and has multiple steps, and it will help any potential user of this codebase to have the steps to reproduce

broken into steps (vasculature synthesis, image synthesis). The authors also provided example simulated data, which is very nice.

Response to reviewer: A revised overview of the process with further detail has been added to L471-L542 and at L822-L828.

- Reviewer comment:** The new experiments that the authors have run and that have been included in the response document are helpful, and we recommend that the authors incorporate them into their manuscript (even if they are included in the supplementary materials). At the moment, I haven't seen any of these new results in their modified manuscript or supplementary materials (if they are included and I have not seen them, apologies for that).

Response to reviewer: Rebuttal figure 1 in the review letter was previously added to the supplemental material (Supplemental Figure 5). The results from the reduced PI-GAN analysis had previously been added to the main manuscript text (L335). In response to Reviewer 2 comments, rebuttal table 2 has now been moved to supplementary material (Supplemental table 3). Rebuttal table 1 was not incorporated into supplementary material because the analysis was included in the main body of the text as described above. Therefore, all new analyses are now incorporated in the manuscript.

- Reviewer comment:** In rebuttal table 2, the mean FID is not necessary; the orders of magnitude between both columns of each row are enough for comparison.

Response to reviewer: The mean FID has been removed from rebuttal table 2 (now Supplemental table 4).

- Reviewer comment:** The model proposed on this paper is complex and I believe a more detailed instruction on how to reproduce each of the steps would be useful.

Response to reviewer: As described above, an extended instruction has been added to the top of the methods section and to the description of code availability.